

# The Community Cloud retrieval for CLimate (CC4CL). Part II: The optimal estimation approach

Gregory R. McGarragh [1], Caroline A. Poulsen [2,3], Gareth E. Thomas [2,3], Adam C. Povey [4], Oliver Sus [5], Stefan Stapelberg [5], Cornelia Schlundt [5], Simon Proud [1], Matthew W. Christensen [1,2,3], Martin Stengel [5], Rainer Hollmann [5], and Roy G. Grainger [4]

[1]Department of Physics, University of Oxford, Clarendon Laboratory, Parks Road, Oxford OX1 3PU, United Kingdom
[2]RAL Space, STFC, Rutherford Appleton Laboratory, Didcot, OX11 0QX, United Kingdom
[3]NCEO, Rutherford Appleton Laboratory, Didcot, OX11 0QX, United Kingdom
[4]National Centre for Earth Observation, Atmospheric, Oceanic and Planetary Physics, University of Oxford, Clarendon Laboratory, Parks Road, Oxford OX1 3PU, United Kingdom
[5]Deutscher Wetterdienst, Frankfurter Str. 135, 63067, Offenbach, Germany

*Correspondence to:* Gregory R. McGarragh (gregory.mcgarragh@physics.ox.ac.uk)

**Abstract.** The Community Cloud retrieval for Climate (CC4CL) is a cloud property retrieval system for satellite-based multispectral imagers and is an important component of the Cloud Climate Change Initiative (Cloud_cci) project. In this paper we discuss the optimal estimation retrieval of cloud optical thickness, effective radius and cloud top pressure based on the Optimal Retrieval of Aerosol and Cloud (ORAC) algorithm. Key to this method is the forward model which, includes the

5    clear-sky model, the liquid water and ice cloud models, the surface model including a bidirectional reflectance distribution function (BRDF), the "fast" radiative transfer solution (which includes a multiple scattering treatment) All of these components and their assumptions and limitations will be discussed in detail. The forward model provides the accuracy appropriate for our retrieval method. The errors are comparable to the instrument noise for cloud optical thicknesses greater than 10. At optical thicknesses less than 10 modelling errors become more significant. The retrieval method is then presented describing

10   optimal estimation in general, the non-linear inversion method employed, measurement and a priori inputs, the propagation of input uncertainties and the calculation of subsidiary quantities that are derived from the retrieval results. An evaluation of the retrieval was performed using measurements simulated with noise levels appropriate for the MODIS instrument. Results show errors less than 10% for cloud optical thicknesses greater than 10. Results for clouds of optical thicknesses less than 10 have errors ranging up to 20%.

## 15   1   Introduction

Remote sensing of clouds from satellites is vitally important for advancing our understanding of the Earth and its climate. Essential cloud parameters to retrieve are optical thickness, particle size and cloud top pressure. These parameters are critical for determining the liquid and ice water content of clouds and for evaluating both radiative and latent heating rates. The methods used to retrieve cloud properties from radiometric measurements made from satellite-based sensors are abound. These





methods differ in the types of measurements used, the assumptions made and in the way the forward problem (the simulation of measurements given cloud properties) is inverted to obtain an estimate of cloud properties from the measurements.

The theoretical basis for retrieving cloud optical thickness and particle size from solar reflectance measurements has been discussed by several authors (Hansen and Pollack, 1970; King, 1997) while measurement studies with airborne sensors have
also been presented (Twomey and Cocks, 1982; Foot, 1988, 1998; Rawlins and Foot, 1990). The retrieval relies on the fact that the reflectance from clouds at non-absorbing visible wavelengths (0.63 and 0.86 µm) is primarily a function of optical thickness whereas in the near-infrared (1.6, 2.1 and 3.7 µm) both liquid water and ice particle absorption become significant so that the reflectance is primarily a function of particle size. Essentially, the problem involves the solution of two non-linear equations for two unknowns. Nakajima and King (1990) formally outlined the theory as it is applied in many retrievals today. In
a companion paper Nakajima et al. (1991) applied the technique to ER-2 based measurements made with a multispectral cloud radiometer at 0.754 and 2.160 µm observing cloud along the same ground track as aircraft-based in situ measurements. The encouraging results were followed by similar retrievals of optical thickness and effective radius using AVHRR (Nakajima and Nakajma, 1995; Ou et al., 1999), MODIS (Rolland et al., 2000; Platnick et al., 2003; King et al., 2010; Platnick et al., 2017), VIIRS (Platnick et al., 2013), GOES (Walther et al., 2013) and SEVIRI (Roebeling et al., 2006). Research has also addressed
the choice of near-infrared channels with several aspects to consider, including stronger particle absorption at 3.7 µm and therefore a stronger function of reflectance to particle size compared to 1.6 and 2.1 µm, deeper photon penetration at 1.6 and 2.1 µm compared to 3.7 µm (Nakajima and King, 1990; Platnick, 2000) and the need to account for both solar and thermal components at 3.7 µm.

At night the retrieval of optical thickness and effective radius is more difficult. Past studies have focused on a split-window
method (Inoue, 1985; Prabhakara et al., 1988; Parol et al., 1991) which relies on the radiative difference of cloud particles at two wavelengths in the infrared window region (8.5–12 µm). Unfortunately, this method is heavily dependent on a priori knowledge of atmospheric temperature and the cloud boundaries, both of which affect the cloud temperature. Studies have shown that explicitly including cloud boundary information from lidar or radar significantly improves retrieval uncertainties (Miller et al., 2000; Cooper et al., 2003) but, of course, this is limited to observations coincident with these active sensors.
The retrieval of cloud top pressure typically relies on matching thermal emission from a cloud in the 11-µm window to a vertical location in a known temperature profile. Without accounting for cloud transparency, this method produces heights that are biased low for semi-transparent clouds due to the contribution of thermal emission from below the cloud. Including additional thermal channels allows for the possibility of retrieving information on cloud transparency from which it is possible to separate the cloud from the below-cloud signal. In particular, the 3.7-µm channel paired with 11 µm has been used to
retrieve cloud top pressure along with a single microphysical parameter that describes the cloud radiative thickness in the infrared, typically referred to as the "effective emissivity" (Szejwach, 1982; Wu, 1987; Liou et al., 1990; Ou et al., 1995). In a similar manor, Ou et al. (1993) has shown that the relative interdependence of optical thickness and effective radius in the 3.7 and 11.0-µm combination allows these to be retrieved along with cloud top pressure, although with a significantly greater uncertainty than using solar wavelengths during the day. $CO_2$-slicing techniques use multiple wavelengths in the 15-µm $CO_2$ band with increasing absorption and therefore increasingly higher peaking weighting functions providing sensitivity




to semi-transparent clouds at a range of heights from the mid to upper troposphere (McCleese and Wilson, 1976; Menzel et al., 1992; Wylie and Menzel, 1999). It is possible to use $CO_2$-slicing results to estimate the cloud top pressure for the thermal methods discussed above to improve the uncertainty in the cloud radiating temperature and the retrieved optical thickness and

effective radius (Cooper et al., 2003). Use of $CO_2$-slicing is limited to instruments that have at least one 15-μm $CO_2$ band, such as MODIS (Menzel et al., 2008) and SEVIRI (Hamann et al., 2014) and is therefore not possible with the long heritage of AVHRR.

Some other retrievals using both solar and thermal channels to obtain information on both optical thickness and/or microphysics and cloud top pressure have been presented. In cases where a near-infrared channel is not available optical thickness and

cloud top pressure may be retrieved with a solar and thermal channel (Rossow and Schiffer, 1991; Minnis et al., 1990, 1993). Alternatively, the radiative thickness can be represented in a "cloud amount" effectively accounting for semi-transparent clouds to obtain a better cloud top pressure (Shenk and Curran, 1973; Reynolds and Vonder Haar, 1977). Finally, Arking and Childs (1985) combined solar (visible and near-infrared) and thermal measurements to estimate optical thickness, a microphysical parameter and cloud top pressure.

The retrieval techniques discussed so far suffer from several drawbacks. First, most of them are separated into solar and thermal methods even though the measurements in these spectral regions are not independent of parameters retrieved in the other. As a result, not all of the available information may be used, i.e. solar information on the thermal optical thickness of semi-transparent clouds and thermal information on particle size. Although some methods discussed above use both solar and thermal channels, their usage is not simultaneous and, therefore, information shared between the different wavelengths may

not be optimally used. In addition, the resulting retrievals may not be radiatively consistent with each other. As a consequence, forward modelling using the solar retrieved optical thickness and effective radius for a cloud with its top placed at the thermally retrieved cloud top pressure may produce simulated radiances that are significantly different than the observed radiances. This inconsistency could have significant impacts on broadband flux computations for radiation studies. Finally, except in some specific cases, these methods tend to lack a formal characterization of their uncertainties which incorporates measurement

noise and the uncertainty of assumed parameters.

The optimal estimation approach to inverse problems is a statistical inversion method based on Bayes' theorem. Application to atmospheric retrievals was presented by Rodgers (1976) and Marks and Rodgers (1993) and formally outlined by Rodgers (2000). Although the method was originally applied to atmospheric temperature and composition retrievals, application to other atmospheric constituents such as clouds began in the late 1990's and has grown steadily in the past 20 years. In optimal

estimation the parameters to be retrieved are inputs into a forward model that produces simulated measurements. These inputs are optimized to obtain the best match between the real and simulated measurements, while being constrained by a priori knowledge of the state. Optimal estimation offers several advantages over more traditional retrieval algorithms:

- It is a universal framework that is able to easily use measurements from different sensors given knowledge of the sensor's noise characteristics. This enables a consistent algorithm with consistent ancillary data sources, minimizing instrument and ancillary data induced changes to the products, improving their utility for studying climate trends.



    – It is able to use any number of channels, where the independent information provided by each channel contributes to the retrieval, maximizing the use of the available information, whereas traditional methods are usually limited to a few pre-selected channels.

– The parameters are retrieved simultaneously, providing a retrieval that is radiatively consistent over the wavelengths of the measurements, provided that the noise characteristics of the instruments are well known.

    – It is able to easily incorporate measurements from multiple sensors for synergistic retrieval algorithms, i.e. passive and active measurements.

    – The same framework may be applied to the retrieval of different parameters such as aerosol and cloud, providing more
consistency in aerosol-cloud interaction studies, for example.

    – A priori information can be explicitly included in the retrieval in a way that is consistent with the measurements. This a priori information can be thought of as virtual measurements that help constrain the retrieval.

    – It provides a rigorous characterization of the retrieval uncertainties, including propagation of measurement noise, the uncertainty of assumed parameters and the uncertainty in the forward model.

– It provides a framework to objectively evaluate the information content of the measurements in a way that is consistent across different measurement sources.

A look at the cloud retrieval literature reveals an increasing usage of optimal estimation including application to AVHRR (Heidinger, 2003; Heidinger and Pavolonis, 2009; Walther and Heidinger, 2012), MODIS (Cooper et al., 2007), ATSR (Poulsen et al., 2012), GOES (Miller et al., 2001; Walther et al., 2013) and SEVIRI (Watts et al., 2011). There have also been several
studies using data from multiple instruments synergistically including passive solar and/or thermal observations combined with cloud profiling radar (Austin and Stephens, 2001; Cooper et al., 2003). Finally, objective information content analyses for cloud retrievals from satellite imagers, using visible, near-infrared and thermal infrared channels simultaneously have been presented (L'Ecuyer et al., 2006; Cooper et al., 2006) including a technique for determining the optimal set of channels using the optimal estimation framework.

This paper is Part II of two papers describing the Community Cloud retrieval for Climate (CC4CL) retrieval system in the context of the Cloud_cci (Hollmann et al., 2013; Stengel et al., 2017) project. Part I (Sus et al., 2017) describes the system as a whole including a description of the supported satellite imagers, ancillary input details, cloud detection and classification, the gridding over space and time, the application of CC4CL to selected scenes, and the validation of those scenes against Cloud-Aerosol Lidar with Orthogonal Polarization (CALIOP) products. Our intention with Part II is to describe the details
of the optimal estimation retrieval including the forward model and the inversion technique based on the Optimal Retrieval of Aerosol and Cloud (ORAC) algorithm (Thomas et al., 2009; Poulsen et al., 2012). Taking advantage of the benefits of an optimal estimation framework ORAC retrieves aerosol, cloud and volcanic ash parameters and supports measurements from several different sensors. The focus of this paper will be on the retrieval of cloud parameters given a set of satellite imager



measurements that are comparatively consistent with the AVHRR heritage channels centered at 0.615/0.630 (second/third generation AVHRR, AVHRR-2/AVHRR-3), 0.862, 1.61 or 3.74, 10.8 and 12.0 μm. Note that for Cloud_cci CC4CL uses this channel configuration even for instruments with additional channels (see Part I for supported instruments) to produce a

consistent time series across sensors, although ORAC, and the CC4CL system in itself, is flexible enough to use any number of imager channels from the visible to the infrared. Finally, a summary of all datasets generated in Cloud_cci using CC4CL is given in Stengel et al. (2017). Section 2 will briefly clarify this, listing the required parameters. Section 3 will present our forward model, discuss assumptions and limitations, and provide a validation using a more advanced, albeit much slower, reference model. Section 4 presents our retrieval method in general, discusses specific details of the use of this method and

some additional products derived from the results while section 5 presents a theoretical study of the performance of the retrieval. Finally, some concluding remarks are given in section 6.

## 2 Data

The method described in this paper requires satellite imager measurements and several ancillary quantities, all of which are prepared for input in a pre-processing stage described in detail in Part I. We will only briefly summarize here what is required.

The measurements include, for each pixel, reflectance for the visible and near-infrared wavelengths and the brightness temperature for the thermal wavelengths. The method also requires the corresponding pixel geolocation (latitude and longitude) and solar and instrument geometry (solar zenith angle, satellite zenith angle and relative azimuth angle, defined as the shortest absolute difference between the solar and satellite azimuth angles). Several ancillary quantities are also required. These include meteorological profiles of pressure, temperature, water vapour and ozone, as well as surface reflectance and emissivity

characteristics. In addition to the measurements and ancillary quantities, an estimate of their uncertainty characteristics is also required for an accurate estimate of the uncertainty of the retrieved quantities. Pre-processing is also responsible for cloud masking and classification and the retrieval methodology described in this paper will assume the properties of liquid water or ice cloud based on the cloud classification.

## 3 Forward model

The forward model contains the physics that simulates radiances as observed by a satellite instrument at the top of the atmosphere (TOA) given both retrieval and assumed model parameters. In addition, derivatives of the TOA radiances with respect to the retrieval parameters must also be computed. The forward model can be thought of as consisting of several component models and a radiative transfer solution that computes the radiances and associated derivatives given the outputs of the component models and solar and instrument geometry. The component models are:

– A clear-sky model including molecular (Rayleigh) scattering, absorption and emission.

– A cloud layer model including cloud particle scattering, absorption and emission.



– A surface reflectance model incorporating ocean and land surface bidirectional reflectance distribution functions (BRDFs).

It is important to develop a forward model that accounts for the physics to the desired accuracy of the retrieval but is also computationally efficient enough to be used for large scale data processing. The ORAC forward model is numerically efficient by making use of both an off-line component and an on-line component. The off-line component handles the expensive particle scattering computations and the multiple scattering radiative transfer computations, the results of which are then used to produce look-up tables (LUTs) of fundamental radiative operators. These LUTs are then used in the on-line component, along with simple arithmetic expressions, to compute the "fast" radiative transfer solution.

## 3.1 Clear-sky model

Our model assumes a plane-parallel atmosphere with the levels defined by the ancillary meteorological input profiles discussed in Part I. The required meteorological inputs are pressure, height, temperature, specific humidity and ozone mixing ratio. The surface is taken to be at the bottom level.

To account for the effects of molecular absorption and emission in a clear-sky atmosphere, transmittances and thermal radiance profiles are required. Specifically, the required transmittances include two profiles, one for transmittance from each level to TOA $[\mathcal{T}_{ac}(p)]$ and one for transmittance from each level to the surface $[\mathcal{T}_{bc}(p)]$, both for a satellite slant path, where $p$ is the atmospheric pressure at a particular level. The required thermal radiance profiles include upwelling radiance, being the total radiance emanating from the layers between the corresponding level and TOA $\left[L_{ac}^{\uparrow}(p)\right]$; downwelling radiance, being the total radiance emanating from the layers between the corresponding level and the surface $\left[L_{ac}^{\downarrow}(p)\right]$; and upwelling radiance, where for each level in the profile is the total radiance from the layers from the surface to that level $\left[L_{bc}^{\uparrow}(p)\right]$.

The transmittances and thermal radiance profiles are computed with the Radiative Transfer for TOVS (RTTOV) model (Saunders et al., 1999; Hocking et al., 2014) version 12.1. RTTOV is a "fast" radiative transfer model for downward-viewing passive visible, infrared and microwave satellite radiometers, spectrometers and interferometers. RTTOV's methodology is based on linear regression of line-by-line computations from LBLRTM version 12.2 (Clough et al., 2005) and the AER molecular database version 3.2. An extensive set of regression predictors are used based on the molecule type with pressure, temperature, specific humidity, ozone mixing ratio and slant path angle as variables. In ORAC, water vapour and ozone are the only variable gases with RTTOV treating effects from other gases using climatology. Additional input parameters include the surface emissivity and the surface skin temperature both discussed in Part I.

In our case RTTOV actually only provides the transmittance $\mathcal{T}_{ac}(p)$ and thermal radiance profiles $L_{ac}^{\uparrow}(p)$ and $L_{ac}^{\downarrow}(p)$ but $\mathcal{T}_{bc}(p)$ and $L_{bc}^{\uparrow}(p)$ can be computed from RTTOV output with

$$\mathcal{T}_{bc}(p) = \mathcal{T}^{*}/\mathcal{T}_{ac}(p) \tag{1}$$

and

$$L_{bc}^{\uparrow}(p) = \left[L_{TOA} - L_{ac}^{\uparrow}(p)\right]/\mathcal{T}_{ac}(p), \tag{2}$$



respectively, where $\mathcal{T}^*$ is the transmittance from the surface to TOA for a satellite slant path and $L_{\mathrm{TOA}}$ is the clear sky TOA radiance.

Since the clear-sky transmittance and emission profiles are independent of cloud, their computation is considered a pre-processing task performed only once in the pre-processing phase discussed in Part I. The effect of the satellite zenith angle is removed from the transmittance profiles with

$$\mathcal{T}_i(0) = \mathcal{T}_i(\overline{\theta_{\mathrm{v}}})^{\cos(\overline{\theta_{\mathrm{v}}})}, \tag{3}$$

where $\mathcal{T}(\theta)$ is transmittance for an air mass factor $1/\cos\theta$ and $\overline{\theta_{\mathrm{v}}}$ is the mean satellite zenith angle within a grid box. An air mass factor is then reapplied to the transmittances on a pixel-by-pixel basis in the RT solution discussed in section 3.6 using the pixel's solar or satellite zenith angles as appropriate. Eq. 3 is an approximation as satellite zenith angle is one of the linear regression predictors, although it has been shown that the effect is minimal compared to other sources of model uncertainty.

In addition to the molecular absorption and emission effects, RTTOV includes an extinction component due to molecular (Rayleigh) scattering in the transmittance calculations. In ORAC this effect must be removed since, as will be discussed in section 3.4, Rayleigh scattering is incorporated into the multiple scattering treatment of the cloud layer. The corrected transmittance $\mathcal{T}_{\mathrm{corr}}$ is given by

$$\mathcal{T}_{\mathrm{corr}}(\lambda, p, \theta_{\mathrm{v}}) = \frac{\mathcal{T}_{\mathrm{RTTOV}}(\lambda, p, \theta_{\mathrm{v}})}{\mathcal{T}_{\mathrm{Ray}}(\lambda, p, \theta_{\mathrm{v}})}, \tag{4}$$

where $\mathcal{T}_{\mathrm{Ray}} = e^{-\tau_{\mathrm{Ray}}(\lambda, p, \theta_{\mathrm{v}})}$, the pressure and path dependent Rayleigh scattering optical thickness $\tau_{\mathrm{Ray}}(\lambda, p, \theta_{\mathrm{v}})$ is

$$\tau_{\mathrm{Ray}}(\lambda, p, \theta_{\mathrm{v}}) = \tau_{\mathrm{Ray}}^0(\lambda)\frac{p}{p_0}\sec\theta_{\mathrm{v}} \tag{5}$$

and the Rayleigh optical thickness of the entire atmosphere at nadir is (Hansen and Travis, 1974)

$$\tau_{\mathrm{Ray}}^0(\lambda) = 0.008569\lambda^{-4}(1 + 0.0113\lambda^{-2} + 0.00013\lambda^{-4}). \tag{6}$$

For each layer bounded by upper and lower pressure levels $p_{\mathrm{u}}$ and $p_{\mathrm{l}}$, respectively, the Rayleigh scattering optical thickness $\tau_{\mathrm{Ray}}(\lambda, p_{\mathrm{l}}, p_{\mathrm{u}})$ that we apply (section 3.4) is computed according to Justus and Paris (1985) with

$$\tau_{\mathrm{Ray}}(\lambda, p_{\mathrm{u}}, p_{\mathrm{l}}) = \tau_{\mathrm{Ray}}(\lambda, p_{\mathrm{u}}) - \tau_{\mathrm{Ray}}(\lambda, p_{\mathrm{l}}), \tag{7}$$

where the Rayleigh optical thickness from TOA to the pressure level $p$ is

$$\tau_{\mathrm{Ray}}(\lambda, p) = \tau_{\mathrm{Ray}}^0(\lambda)\exp(-0.1188p - 0.00116p^2) \tag{8}$$

and the Rayleigh optical thickness of the entire atmosphere is

$$\tau_{\mathrm{Ray}}^0(\lambda) = \frac{p_{\mathrm{s}}}{p_0}\frac{1}{115.6406\lambda^4 - 1.335\lambda^2}, \tag{9}$$

where $p_0 = 1013.25$ hPa, $p_{\mathrm{s}}$ is the surface pressure in hPa and $\lambda$ is in μm. The Rayleigh single scattering albedo is implicitly set to unity and the single scattering phase function $P_{\mathrm{Ray}}(\lambda, \Theta)$ is computed with

$$P_{\mathrm{Ray}}(\lambda, \Theta) = \frac{3}{4}\left(1 + \cos^2\Theta\right) + \left(1 - \frac{1 - \delta}{1 + \delta/2}\right), \tag{10}$$





where the depolarization factor $\delta$ is taken to be constant at 0.0279 from Young (1981).

Note that, ORAC does not take the variation of surface pressure due to terrain height or meteorology into account. Combined with the lack of polarization in the radiative transfer calculations, this means that ORAC is not currently suitable for use with instrument channels in the blue or ultraviolet, where the Rayleigh signal is much stronger and will vary significantly with terrain height. The effects due to polarization at these small wavelengths would require a full vector radiative transfer solution.

## 3.2   Cloud layer model

The cloud layer model accounts for particle scattering, absorption and emission effects and is parameterized in terms of particle type (liquid water droplets or ice crystals) and the following retrieved quantities: cloud optical thickness at 0.55 μm $\tau_{0.55,\mathrm{c}}$, effective radius of the cloud particle size distribution $r_{\mathrm{e,c}}$ and cloud top pressure $p_{\mathrm{c}}$.

The cloud layer model is assumed to consist of a single layer containing only a single particle type. The cloud is assumed

to be geometrically infinitely thin and plane-parallel and is linearly interpolated into the plane-parallel atmospheric model at a given cloud top pressure $p_{\mathrm{c}}$ producing values of pressure, height and temperature for the cloud at $p_{\mathrm{c}}$. In addition, transmittances $\mathcal{T}_{\mathrm{ac}}(p_{\mathrm{c}})$ and $\mathcal{T}_{\mathrm{bc}}(p_{\mathrm{c}})$, and thermal radiance quantities $L_{\mathrm{ac}}^{\uparrow}(p_{\mathrm{c}})$, $L_{\mathrm{ac}}^{\downarrow}(p_{\mathrm{c}})$ and $L_{\mathrm{bc}}^{\uparrow}(p_{\mathrm{c}})$ are also interpolated from their respective profiles discussed in section 3.1. An infinitely thin cloud model allows for a significant performance gain as the effects of the atmospheric gases are separated from that of the cloud. This is an important part of the "fast" radiative transfer solution

discussed in section 3.6.

Optical thickness $\tau(\lambda)$ is defined as:

$$\tau(\lambda) = \int_0^H \beta_{\mathrm{e}}(z,\lambda)\,dz = \int_0^H (\beta_{\mathrm{s}} + \beta_{\mathrm{a}})(z,\lambda)\,dz, \tag{11}$$

where $z$ is the vertical depth into the cloud, $H$ is the geometric thickness of the cloud, $\lambda$ is wavelength and $\beta_{\mathrm{e}}(\lambda)$ is the extinction coefficient defined as the sum of the scattering and absorption components, $\beta_{\mathrm{s}}(\lambda)$ and $\beta_{\mathrm{a}}(\lambda)$, respectively. The

extinction coefficient $\beta_{\mathrm{e}}(\lambda)$ may be related to the extinction cross section $\sigma_{\mathrm{e}}(\lambda,r)$ and efficiency $Q_{\mathrm{e}}(\lambda,r)$ by

$$\beta_{\mathrm{e}}(\lambda) = \int_0^\infty \sigma_{\mathrm{e}}(\lambda,r)n(r)\,dr = \int_0^\infty \pi r^2 Q_{\mathrm{e}}(\lambda,r)n(r)\,dr, \tag{12}$$

where $r$ is, in general, the sphere equivalent particle radius, $n(r)$ is the particle size distribution in terms of number density and $\sigma_{\mathrm{e}}(\lambda) = \sigma_{\mathrm{s}}(\lambda) + \sigma_{\mathrm{a}}(\lambda)$ and $Q_{\mathrm{e}}(\lambda) = Q_{\mathrm{s}}(\lambda) + Q_{\mathrm{a}}(\lambda)$, are sums of their scattering and absorption components, respectively.

If we introduce the normalized size distribution $\hat{n}(r) = n(r)/N$, where

$$N = \int_0^\infty n(r)\,dr, \tag{13}$$

5   then Eq. 12 may be written as

$$\beta_{\mathrm{e}}(\lambda) = N\hat{\sigma}_{\mathrm{e}}(\lambda) = N \int_0^\infty \sigma_{\mathrm{e}}(\lambda,r)\hat{n}(r)\,dr = N \int_0^\infty \pi r^2 Q_{\mathrm{e}}(\lambda,r)\hat{n}(r)\,dr, \tag{14}$$





where $\hat{\sigma}_{\mathrm{e}}(\lambda)$ is the normalized ensemble-averaged extinction cross section. This value is dependent on the shape of the size distribution but independent of the density of particles and represents the spectrally dependent microphysical influence on the cloud optical thickness. Since the cloud is parameterized according to the cloud optical thickness at 0.55 μm but forward model simulations must be performed in each wavelength channel the required optical thickness $\tau(\lambda)$ in these channels must be determined from the spectral variation defined by

$$\tau(\lambda) = \frac{\hat{\sigma}_{\mathrm{e}}(\lambda)}{\hat{\sigma}_{\mathrm{e}}(0.55)}\tau(0.55). \tag{15}$$

The effective radius $r_{\mathrm{e}}$ is defined as the ratio of the 3rd and 2nd moments of the particle size distribution:

$$r_{\mathrm{e}} = \frac{\int_0^\infty r^3 n(r)\,dr}{\int_0^\infty r^2 n(r)\,dr}, \tag{16}$$

while the effective variance of the distribution is defined as:

$$v_{\mathrm{e}} = \frac{\int_0^\infty (r - r_{\mathrm{e}})^2 r^2 n(r)\,dr}{\int_0^\infty r^2 n(r)\,dr}. \tag{17}$$

Hansen and Travis (1974) and Mishchenko and Travis (1994) show that different size distributions that have the same effective radius and effective variance have similar scattering and absorption properties. Thus by parameterizing the size distribution in terms of effective radius we reduce the dependency on the details of the size distribution depending mainly on the assumption of the distribution's width.

It is from these relationships and further assumptions based on particle type that the three fundamental radiative transfer quantities required for the multiple scattering computations discussed in section 3.4 are parameterized: the total optical thickness $\tau(\lambda)$, the ensemble-averaged single scattering albedo $\omega(\lambda)$ and the ensemble-averaged single scattering phase matrix $\mathbf{F}(\lambda, \Theta)$. It is through these parameterizations that the radiative transfer quantities depend on particle concentration and size distribution.

The single scattering albedo $\omega(\lambda)$ is the proportion of radiation incident on a particle that is scattered rather than absorbed and is given as

$$\omega(\lambda) = \frac{\beta_{\mathrm{s}}(\lambda)}{\beta_{\mathrm{e}}(\lambda)} = \frac{\hat{\sigma}_{\mathrm{s}}(\lambda)}{\hat{\sigma}_{\mathrm{e}}(\lambda)}. \tag{18}$$

The $(4 \times 4)$ single scattering phase matrix $\mathbf{F}(\lambda, \Theta)$ describes the angular distribution and state of polarization of scattered radiation given the state of polarization of the incident radiation, where the radiation is described by the 4-element Stokes vector $\boldsymbol{I} = [I, Q, U, V]$. This form of the phase matrix, depending only on the angle between the incident and scattering directions $\Theta$, is valid for mediums composed of randomly oriented particles. The ORAC forward model makes the so-called "scalar approximation" where only the intensity $I$ is considered. In this case only the (1,1) element of the phase matrix is required which is referred to as the single scattering phase function $P(\lambda, \Theta)$ with the following normalization condition

$$\frac{1}{2} \int_0^\pi P(\lambda, \Theta) \sin \Theta \, d\Theta = 1. \tag{19}$$





Like $\hat{\sigma}_{\mathrm{e}}(\lambda)$, both $\omega(\lambda)$ and $P(\lambda, \Theta)$ are size distribution normalized so they are independent of the density of particles.

The equations presented so far in this section are independent of particle type and shape and are valid for all mediums of randomly oriented particles. It is the source of the normalized scattering and absorption cross sections $\hat{\sigma}_{\mathrm{s}}(\lambda, r)$ and $\hat{\sigma}_{\mathrm{a}}(\lambda, r)$ and the phase function $P(\lambda, \Theta)$ that varies with particle type. They are computed with physical models such as Mie theory (Mie, 1908) or T-matrix theory (Mishchenko et al., 2002), or in the case of ice crystals, geometric optics (Liou, 2002).

### 3.2.1 Liquid water droplets

For liquid water droplets the Mie theory code implementation presented by Grainger et al. (2004) is used for the computation of $Q_{\mathrm{s}}[\lambda, r, m(\lambda)]$, $Q_{\mathrm{a}}[\lambda, r, m(\lambda)]$ and $P[\lambda, r, m(\lambda), \Theta]$, where $m(\lambda) = m_{\mathrm{r}}(\lambda) + im_{\mathrm{i}}(\lambda)$ is the refractive index of the particle composition. It is convenient to integrate the computations across an analytical size distribution described by a limited number of parameters for which we use the modified gamma distribution given by Deirmendjian (1969). The distribution, in terms of number density as a function of radius $n(r)$ is given as

$$n(r) = ar^{\alpha} \exp(-br^{\gamma}), \tag{20}$$

where the parameters $a$, $\alpha$, $b$ and $\gamma$ are positive, $\alpha$ is an integer and the mode $r_{\mathrm{m}}$ of the distribution occurs when $r = \left(\frac{\alpha}{b\gamma}\right)^{1/\gamma}$. In ORAC $a = 2.373$, $\alpha = 6$, $b = \alpha/r_{\mathrm{m}}$ and $\gamma = 1$.

As discussed, the size distribution is parameterized in terms of effective radius which is related to the modified-gamma distribution of Eq. 20 by

$$r_{\mathrm{e}} = \frac{r_{\mathrm{m}}(\alpha + 3)}{\alpha}, \tag{21}$$

while the effective variance is related to the modified-gamma distribution by

$$v_{\mathrm{e}} = \frac{1}{\alpha + 3}. \tag{22}$$

For the complex index of refraction for liquid water, values were taken from Hale and Querry (1973) for wavelengths in the range $0.25 \leq \lambda \leq 0.69$ µm, Palmer and Williams (1974) for $0.69 < \lambda \leq 2.0$ µm, and Downing and Williams (1975) for $\lambda > 2.0$ µm.

### 3.2.2 Ice crystals

Ice crystal single scattering property models provided by Baum et al. (2011, 2014) are used, which are based on in situ cloud microphysical measurements and single-scattering calculations. The in situ measurements provide information on particle size distributions, ice water content and the median mass diameter (Heymsfield et al., 2013). The single scatter properties are provided by Yang et al. (2013) and are based on a combination of the Amsterdam Discrete Dipole Approximation, the T-matrix method and the improved geometric-optics method. They are provided for each habit at 189 discrete sizes between 2 and 10,000 µm and for 445 discrete wavelengths ranging from 0.2 to 100 µm.





The bulk single scattering properties are computed for each wavelength by integrating over particle size distribution and the nine ice particle habits to produce size distribution averaged properties for a "general habit mixture". These are made available as a function of effective radius in 23 bins from 5 to 60 μm. It has been shown that ice particle roughening significantly impacts the single scattering phase function (Baum et al., 2010) and that the general habit mixture with severe roughening provides the closest comparison with polarization measurements (Cole et al., 2013). These are the models used in the ORAC retrieval.

As will be shown later, the retrieval requires ice crystal scattering properties up to an effective radius of 92 μm. The ice crystal properties are only provided up to 60 μm and therefore linear extrapolation is used up to 92 μm. The error incurred in the approximation is minimal as the variation in the properties tends to flatten as effective radius increases.

### 3.3 Surface reflectance model

The surface is characterized by a BRDF which is computed differently for ocean and land surface. The BRDF over ocean is computed using the methodology outlined by Sayer et al. (2010) which includes 3 components:

$$\rho_{\text{ocean}}(\lambda, \theta_0, \theta_{\text{v}}, \Delta\phi, u, v)\rho_{\text{sg}}(\lambda, \theta_0, \theta_{\text{v}}, \Delta\phi, u, v) + \rho_{\text{wc}}(\lambda, u, v) + \rho_{\text{ul}}(\lambda, \theta_0, \theta_{\text{v}}, C), \tag{23}$$

where $\rho_{\text{sg}}$ is the sun-glint off wave facets (Cox and Munk, 1954), $\rho_{\text{wc}}$ is the reflectance from surface foam, so-called "white-caps" (Koepke, 1984), and $\rho_{\text{ul}}$ is the scattering from within the water, so-called "underlight" (Morel and Prieur, 1977). The required physical parameters include the horizontal wind vector $u$ and $v$ (m s$^{-1}$), obtained from the meteorological input, to determine wave statistics and white-cap coverage, as well as chlorophyll and dissolved organic matter (CDOM) concentration $C$ (mg m$^{-3}$), a globally averaged value obtained from climatology.

The BRDF over land is a weighted sum of an isotropic kernel (unity) and 2 BRDF kernels (Wanner et al., 1997), both dependent on solar and satellite geometry only:

$$\rho_{\text{land}}(\lambda, \theta_0, \theta_{\text{v}}, \Delta\phi) = f_{\text{iso}}(\lambda) + f_{\text{vol}}(\lambda)K_{\text{vol}}(\theta_0, \theta_{\text{v}}, \Delta\phi) + f_{\text{geo}}(\lambda)K_{\text{geo}}(\theta_0, \theta_{\text{v}}, \Delta\phi), \tag{24}$$

where $K_{\text{vol}}(\theta_0, \theta_{\text{v}}, \Delta\phi)$ is known as the the Ross-thick kernel which parameterizes volumetric scattering and $K_{\text{geo}}(\theta_0, \theta_{\text{v}}, \Delta\phi)$ is the Li-sparse kernel which parameterizes geometric shadowing. Although our method is independent of the source of the weights $f(\lambda)$, in the current CC4CL implementation the weights are provided by the 0.05° MODIS MCD43C1 BRDF ancillary input discussed along with references in Part I.

Over snow and ice the surface reflectance is assumed to be Lambertian with reflectance values taken from the The ASTER Spectral Library Version 2.0 (Baldridge et al., 2009). This reflectance is combined with the reflectance for either ocean or land as described above based on the fraction of snow/ice provided by the pre-processing stage described in Part I.

### 3.4 Reflectance and transmission operators

The next step in the forward model is the computation of reflectance, transmission and emissivity operators which are used in the "fast" RT solution described in section 3.6. This computation involves the solution to the radiative transfer equation (RTE)



for monochromatic radiation through a single plane-parallel homogeneous layer given as

$$\mu \frac{dL(\lambda, \tau, \mu, \phi)}{d\tau(\lambda)} = L(\lambda, \tau, \mu, \phi) - J(\lambda, \tau, \mu, \phi), \tag{25}$$

where $L(\lambda, \tau, \mu, \phi)$ is the radiance along the direction specified by the cosine of the polar angle $\mu$ and the azimuthal angle $\phi$ at optical depth $\tau(\lambda)$ measured perpendicular to the surface of the medium. The second term on the right hand side is the source function given by

$$J(\lambda, \tau, \mu, \phi) = \frac{\omega(\lambda)}{4\pi} \int\limits_{-1}^{1} \int\limits_{-1}^{1} P(\lambda, \mu, \phi, \mu', \phi') L(\lambda, \tau, \mu', \phi') d\mu' d\phi' - S_{\text{solar}}(\lambda, \tau, \mu, \phi) - S_{\text{thermal}}(\lambda), \tag{26}$$

where $P(\lambda, \mu, \phi, \mu', \phi')$ is the single scattering phase function for radiation from the direction $(\mu', \phi')$ scattered into the direction $(\mu, \phi)$ and $\omega(\lambda)$ is the single scattering albedo. The first term on right hand side of Eq. 26 represents the source from multiple scattering. The second term is the solar single scattering source given by

$$S_{\text{solar}}(\lambda, \tau, \mu, \phi) = \frac{\omega(\lambda)}{4\pi} E_0(\lambda) P(\lambda, \mu, \phi, -\mu_0, \phi_0) e^{-\tau(\lambda)/\mu_0}, \tag{27}$$

where $E_0(\lambda)$ is the TOA incident solar irradiance, and $\mu_0$ and $\phi_0$ are the solar zenith and azimuth angles, respectively. The third term on the right hand side of Eq. 26 is the thermal emission source given by

$$S_{\text{thermal}}(\lambda, T) = [1 - \omega(\lambda)] B(\lambda, T), \tag{28}$$

where $B(\lambda, T)$ is the Planck black body function and $T$ is the average layer temperature.

For performance reasons, the operators are precomputed and stored in an LUT from which the values for an arbitrary set of geometric and optical parameters may be interpolated. The LUTs are computed with the DIscrete Ordinates Radiative Transfer (DISORT) software package (Stamnes et al., 1988). This step, although slow, is performed off-line and the resulting look-up LUTs are static. DISORT is a general purpose plane-parallel RTE solver for monochromatic radiation with support for multiple layers, absorption and multiple scattering, and solar and thermal emission sources. It implements the delta-M correction of Wiscombe (1977) for strongly asymmetric phase functions and the TMS single scattering correction of Nakajima and Tanaka (1988).

DISORT still makes approximations, which can limit its accuracy in certain circumstances. The most important of these are:

– It assumes a plane-parallel atmosphere, so does not account for the curvature of the Earth. This is important at solar and satellite zenith angles greater than approximately $75°$.

– It is a one dimensional model, so cannot reproduce the effects of horizontal gradients in the scattering medium. This is important where strong gradients exist, such as near cloud edges and/or in broken cloud fields, but this limitation is not relevant as we treat each pixel independently.

– It does not model polarization effects and so cannot be used to model measurements made by instruments which are sensitive to polarization. In addition, this so-called "scalar approximation" does not take into account polarization introduced





**Table 1.** The dimensions of the ORAC LUTs used in the liquid water cloud retrieval. Note that not all LUTs are functions of all variables (for instance, atmospheric transmission terms are functions of a single zenith angle only).

| Parameter | No. points | Min. value | Max. value | Spacing |
|---|---|---|---|---|
| $\tau_{0.55,\mathrm{c}}$ | 18 | $1.0 \times 10^{-20}$ | 256.0 | $\log_{10}$ |
| $r_{\mathrm{e,c}}$ (μm) | 20 | 1.0 | 39.0 | linear |
| $\theta_0$ | 10 | 0° | 89° | linear |
| $\theta_\mathrm{v}$ | 10 | 0° | 89° | linear |
| $\Delta\phi$ | 11 | 0° | 180° | linear |

5    into the diffuse component of radiance by Rayleigh scattering and/or the surface, and the subsequent depolarization effect of particles.

To compute the operators DISORT must be provided with solar and instrument geometry and the optical thickness, single scattering albedo and phase function for each layer. In addition to particle effects, the LUTs account for Rayleigh scattering and therefore, even though the operators are for a single homogeneous cloud, the computation is performed for an entire

10    atmospheric profile. For this we use the mid-latitude summer profile provided by McClatchey et al. (1972) to compute the Rayleigh scattering optical properties in each layer as described in section 3.1. In order to account for multiple scattering between molecules and cloud particles we give the cloud layer a physical thickness of 1 km and place it at a top pressure of 560 hPa. In this layer, the optical properties for Rayleigh scattering and those of the cloud, must be combined by (assuming dependence on $\lambda$):

$$\tau = \tau_{\mathrm{Ray}} + \tau_{\mathrm{c}}, \tag{29}$$

$$\omega = \frac{\tau_{\mathrm{Ray}} + \omega_{\mathrm{c}}\tau_{\mathrm{c}}}{\tau_{\mathrm{R}} + \tau_{\mathrm{c}}}, \tag{30}$$

$$P(\theta) = \frac{\tau_{\mathrm{Ray}}P_{\mathrm{Ray}}(\theta) + \tau_{\mathrm{c}}\omega_{\mathrm{c}}P_{\mathrm{c}}(\theta)}{\tau_{\mathrm{Ray}} + \tau_{\mathrm{c}}\omega_{\mathrm{c}}}, \tag{31}$$

where the Rayleigh single scattering albedo is equal to unity. The computations are performed at optical thickness, effective

5    radius, solar zenith angle, satellite zenith angle and relative azimuth angle vertices defined by the dimensions in Tables 1 and 2 for liquid water and ice cloud, respectively.

The reflectance and transmission operators represent the transfer of either direct beam or diffuse incoming radiation resulting in either direct beam or diffuse outgoing radiation. They are computed separately for both direct beam and diffuse incoming sources with results for both direct beam or diffuse outgoing radiation being produced simultaneously. In addition, an operator

10    representing the emissivity of the cloud is produced by including a thermal source in the cloud layer. The calculations are performed mono-chromatically at the central wavelength for each instrument channel. The error incurred by not accounting for the instrument channel response function has been shown to be small compared to other sources of uncertainty in the forward model. A total of seven operators for each channel are required (assuming dependence on $\tau_{0.55,\mathrm{c}}$ and $r_{\mathrm{e,c}}$):





**Table 2.** The dimensions of the ORAC LUTs used in the ice cloud retrieval. Note that not all LUTs are functions of all variables (for instance, atmospheric transmission terms are functions of a single zenith angle only).

| Parameter | No. points | Min. value | Max. value | Spacing |
|---|---|---|---|---|
| $\tau_{0.55,c}$ | 18 | $1.0 \times 10^{-20}$ | 256.0 | $\log_{10}$ |
| $r_{e,c}$ (μm) | 23 | 4.0 | 92.0 | linear |
| $\theta_0$ | 10 | $0°$ | $89°$ | linear |
| $\theta_v$ | 10 | $0°$ | $89°$ | linear |
| $\Delta\phi$ | 11 | $0°$ | $180°$ | linear |

1. $R_{bb}(\lambda, \theta_0, \theta_v, \Delta\phi)$: The bidirectional reflectance of the cloud.

2. $R_{db}(\lambda, \theta_v)$: The downward diffuse reflectance from the cloud, as viewed from a specific direction.

3. $R_{dd}(\lambda)$: The bihemispherical reflectance of the cloud.

4. $\varepsilon(\lambda, \theta_v)$: The emissivity of the cloud, as viewed from a specific direction.

5. $T_{bb}^{\downarrow}(\lambda, \theta_0)$: The downward direct transmission of the cloud of the direct solar beam.

6. $T_{bb}^{\uparrow}(\lambda, \theta_v)$: The upward direct transmission of the cloud into the viewing direction.

7. $T_{bd}^{\downarrow}(\lambda, \theta_0)$: The downward diffuse transmission of the cloud, as illuminated by the direct solar beam.

8. $T_{db}^{\uparrow}(\lambda, \theta_v)$: The upward diffuse transmission of the cloud, as viewed from a specific direction.

A schematic diagram of the operators and their interaction with the surface is presented in Figure 1. A $\downarrow$ denotes transmission from the top to the bottom of the atmosphere, while a $\uparrow$ indicates the reverse. Dependence on the solar zenith, satellite zenith and relative azimuth angles are denoted by $\theta_0$, $\theta_v$ and $\Delta\phi$ respectively. The pairs of b and d subscripts denote the type of radiation each term operates on and produces; for example, $T_{bd}^{\downarrow}(\lambda, \theta_0)$ operates on the direct beam (b) of solar radiation and produces the diffuse radiation (d) that results at the bottom of the atmosphere. Note that $T_{bb}^{\downarrow}(\lambda, \theta_0) \equiv T_{bb}^{\uparrow}(\lambda, \theta_v)$ when $\theta_0 = \theta_v$ so that a single LUT can be used for each of the these for a total of seven LUTs.

The inclusion of Rayleigh scattering effects in the reflectance and transmission operators requires some approximation. This is because the Rayleigh scattering parameters included are defined for each individual atmospheric layer whereas the cloud layer is added to a single layer. This cloud layer must be placed at a fixed top pressure within the atmosphere (560 hPa) when producing the LUTs, since the cloud top pressure is not an LUT variable. This is an approximation since Rayleigh scattering effects are pressure dependent and therefore vary with height. This means that when the retrieval places the particle layer higher than 560 hPa, the Rayleigh scattering effects will be slightly over estimated whereas if it is placed lower than





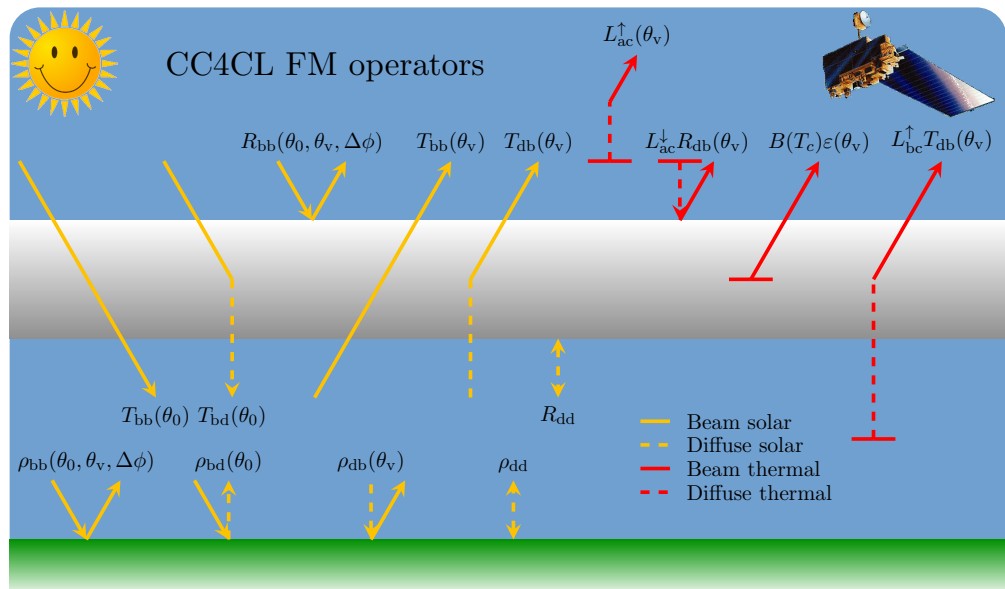

**Figure 1.** A schematic diagram of the ORAC cloud reflectance, transmission and emission operators; surface reflectance operators; and clearsky emission profiles. Yellow arrows indicate reflectance and transmission operators and red arrows indicate emission. Solid lines indicate beam transport and the dashed lines indicate diffuse transport.

560 hPa, the Rayleigh scattering will be slightly underestimated. Rayleigh scattering could of course be included in the clearsky transmittances but this would be an extinction only effect and would not account for multiple scattering between molecules and between molecules and particles.

### 3.5 Surface reflectance operators

Interaction with the surface is parameterized by four reflectance operators (assuming dependence on the appropriate BRDF kernel parameters discussed in section 3.3):

1. $\rho_{\mathrm{bb}}(\lambda, \theta_0, \theta_\mathrm{v}, \Delta\phi)$: The bidirectional reflectance. This is the reflectance of the surface to direct beam illumination at $\theta_0$, as viewed from a specific direction $\theta_\mathrm{v}$. It is the reflectance that would be observed by a satellite instrument in the absence of an atmosphere.

2. $\rho_{\mathrm{bd}}(\lambda, \theta_0)$: The directional-hemispheric reflectance. This is the fraction of incoming direct beam illumination at $\theta_0$ that is reflected across all zenith angles. This is also referred to as the *black-sky albedo*.

3. $\rho_{\mathrm{db}}(\lambda, \theta_\mathrm{v})$: The hemispheric-directional reflectance. This is the reflectance of the surface to purely diffuse illumination, as viewed from a specific direction $\theta_\mathrm{v}$.





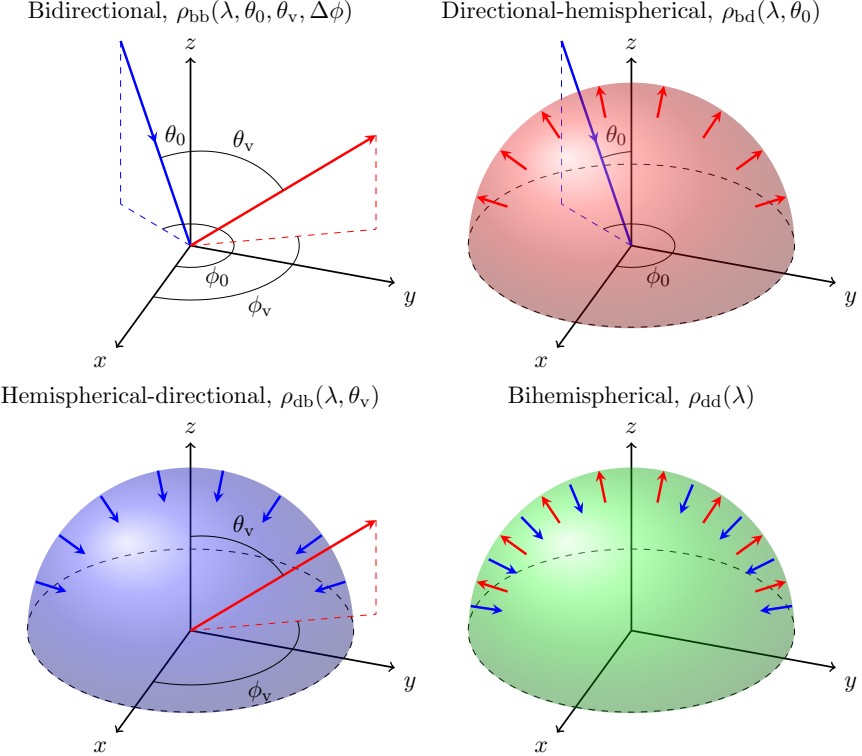

**Figure 2.** Schematic diagrams of each of the four ORAC surface reflectance operators. Blue represents incident radiation and red represents reflected radiation. The long arrows represent beam transport and the hemispheres with short arrows represent diffuse transport.

4. $\rho_{\mathrm{dd}}(\lambda)$: The bihemispherical reflectance. This is the reflectance of the surface to purely diffuse illumination, across all viewing directions. This is also referred to as the *white-sky albedo*.

A schematic diagram of the operators and their interaction with a cloud is presented in Figure 1 while a more detailed diagram of the operators themselves is given in Figure 2.

The first term $\rho_{\mathrm{bb}}(\lambda, \theta_0, \theta_{\mathrm{v}}, \Delta\phi)$ is computed directly from the BRDF over land or ocean, where $\rho_{\mathrm{bb}}(\lambda, \theta_0, \theta_{\mathrm{v}}, \Delta\phi) =$

5 $\rho_{\mathrm{ocean}}(\lambda, \theta_0, \theta_{\mathrm{v}}, \Delta\phi)$ or $\rho_{\mathrm{bb}}(\lambda, \theta_0, \theta_{\mathrm{v}}, \Delta\phi) = \rho_{\mathrm{land}}(\lambda, \theta_0, \theta_{\mathrm{v}}, \Delta\phi)$ over ocean or land, respectively. The three other terms are derived from the BRDF integrated over solar and/or satellite geometry written as (assuming dependence on $\lambda$)

$$\rho_{\mathrm{bd}}(\theta_0) = \frac{\int_0^{2\pi} \int_0^{\pi/2} \rho_{\mathrm{bb}}(\theta_0, \theta_{\mathrm{v}}, \Delta\phi) \cos\theta_{\mathrm{v}} \sin\theta_{\mathrm{v}} \, d\theta_{\mathrm{v}} \, d\Delta\phi}{\int_0^{2\pi} \int_0^{\pi/2} \cos\theta_{\mathrm{v}} \sin\theta_{\mathrm{v}} \mathrm{d}\theta_{\mathrm{v}} \mathrm{d}\Delta\phi}$$

$$= \frac{1}{\pi} \int\limits_0^{2\pi} \int\limits_0^{\pi/2} \rho_{\mathrm{bb}}(\theta_0, \theta_{\mathrm{v}}, \Delta\phi) \cos\theta_{\mathrm{v}} \sin\theta_{\mathrm{v}} \, d\theta_{\mathrm{v}} \, d\Delta\phi, \tag{32}$$


$$\rho_{\mathrm{db}}(\theta_{\mathrm{v}}) = \frac{\int_0^{2\pi}\int_0^{\pi/2} \rho_{\mathrm{bb}}(\theta_0,\theta_{\mathrm{v}},\Delta\phi)\cos\theta_0\sin\theta_0\,d\theta_0\,d\Delta\phi}{\int_0^{2\pi}\int_0^{\pi/2}\cos\theta_0\sin\theta_0\mathrm{d}\theta_0\mathrm{d}\Delta\phi}$$

$$= \frac{1}{\pi}\int\limits_0^{2\pi}\int\limits_0^{\pi/2}\rho_{\mathrm{bb}}(\theta_0,\theta_{\mathrm{v}},\Delta\phi)\cos\theta_0\sin\theta_0\,d\theta_0\,d\Delta\phi \tag{33}$$

and

$$\rho_{\mathrm{dd}} = \frac{\int_0^{\pi/2}\rho_{\mathrm{bd}}(\theta_0)\cos\theta_0\sin\theta_0\mathrm{d}\theta_0}{\int_0^{\pi/2}\cos\theta_0\sin\theta_0\,d\theta_0}$$

$$= 2\int\limits_0^{\pi/2}\rho_{\mathrm{bd}}(\theta_0)\cos\theta_0\sin\theta_0\,d\theta_0. \tag{34}$$

Note that $\rho_{\mathrm{db}}(\lambda,\theta_0) \equiv \rho_{\mathrm{bd}}(\lambda,\theta_{\mathrm{v}})$ when $\theta_0 = \theta_v$, that is they are both hemispherical reflectance, just seen from different zenith angles.

### 3.6 The "fast" radiative transfer solution

The "fast" radiative transfer solution uses the cloud reflectance, cloud transmission and surface reflectance operators along with clear-sky transmittances to simulate the measurements as observed by a satellite sensor at TOA. Shortwave solar reflectance computations are separated from the longwave thermal infrared brightness computations. Although not strictly required, this separation results in an efficient implementation as components such as surface reflectance or cloud top emission are specific to solar and thermal wavelengths, respectively. At thermal wavelengths, where there is a significant solar contribution, separate calculations are performed and the resulting solar reflectance is converted to brightness temperature and added to the resulting thermal brightness temperature computation.

#### 3.6.1 Solar reflectance

Using the reflectance and transmission operators described in section 3.4, the surface reflectance operators described in section 3.5, and neglecting molecular absorption, the observed reflectance of the cloud/surface system can be written as (assuming



dependence on wavelength $\lambda$):

$$
\begin{aligned}
R(\theta_0, \theta_v, \Delta\phi) = \ & R_{bb}(\theta_0, \theta_v, \Delta\phi) \\
& + T_{bb}^{\downarrow}(\theta_0)\rho_{bb}(\theta_0, \theta_v, \Delta\phi)T_{bb}^{\uparrow}(\theta_v) \\
& + T_{bb}^{\downarrow}(\theta_0)\rho_{bd}(\theta_0)T_{db}^{\uparrow}(\theta_v) \\
& + T_{bd}^{\downarrow}(\theta_0)\rho_{db}(\theta_v)T_{bb}^{\uparrow}(\theta_v) \\
& + T_{bd}^{\downarrow}(\theta_0)\rho_{dd}T_{db}^{\uparrow}(\theta_v) \\
& + T_{bb}^{\downarrow}(\theta_0)\rho_{bd}(\theta_0)R_{dd}\rho_{db}T_{bb}^{\uparrow}(\theta_v) \\
& + T_{bb}^{\downarrow}(\theta_0)\rho_{bd}(\theta_0)R_{dd}\rho_{dd}T_{db}^{\uparrow}(\theta_v) \\
& + T_{bd}^{\downarrow}(\theta_0)\rho_{dd}R_{dd}\rho_{db}T_{bb}^{\uparrow}(\theta_v) \\
& + T_{bd}^{\downarrow}(\theta_0)\rho_{dd}R_{dd}\rho_{dd}T_{db}^{\uparrow}(\theta_v) \\
& + T_{bb}^{\downarrow}(\theta_0)\rho_{bd}(\theta_0)R_{dd}\rho_{dd}R_{dd}\rho_{db}T_{bb}^{\uparrow}(\theta_v) \\
& + T_{bb}^{\downarrow}(\theta_0)\rho_{bd}(\theta_0)R_{dd}\rho_{dd}R_{dd}\rho_{dd}T_{db}^{\uparrow}(\theta_v) \\
& + T_{bd}^{\downarrow}(\theta_0)\rho_{dd}R_{dd}\rho_{dd}R_{dd}\rho_{db}T_{bb}^{\uparrow}(\theta_v) \\
& + T_{bd}^{\downarrow}(\theta_0)\rho_{dd}R_{dd}\rho_{dd}R_{dd}\rho_{dd}T_{db}^{\uparrow}(\theta_v) \\
& + \ldots
\end{aligned}
\tag{35}
$$

Here we have four terms resulting from a single surface reflection in Eq. 35, which can be described as follows:

1. $T_{bb}^{\downarrow}(\theta_0)\rho_{bb}(\theta_0, \theta_v, \Delta\phi)T_{bb}^{\uparrow}(\theta_v)$ is the directly transmitted solar beam that is reflected off the surface into the viewing direction of the satellite and directly transmitted back through the atmosphere.

2. $T_{bb}^{\downarrow}(\theta_0)\rho_{bd}(\theta_0)T_{db}^{\uparrow}(\theta_v)$ is the directly transmitted solar beam, that is diffusely reflected off the surface and diffusely transmitted into the viewing direction of the satellite.

3. $T_{bd}^{\downarrow}(\theta_0)\rho_{db}(\theta_v)T_{bb}^{\uparrow}(\theta_v)$ is the diffusely transmitted solar beam, that is reflected off the surface into the viewing direction of the satellite and directly transmitted back through the atmosphere.

4. $T_{bd}^{\downarrow}(\theta_0)\rho_{dd}T_{db}^{\uparrow}(\theta_v)$ is the diffusely transmitted solar beam, that is diffusely reflected off the surface and diffusely transmitted into the viewing direction of the satellite.

The terms following on from these describe the rapidly diminishing series of multiple reflections between the surface and overlaying atmosphere. For these terms the assumption has been made that the surface/cloud pair are essentially Lambertian reflectors so that only the bihemispherical reflectance of the cloud is required in the series. This is equivalent to saying, that neglecting directly transmitted solar radiation, the sky is equally bright in all directions.





By gathering terms, Eq. 35 can be simplified to give

$$
\begin{aligned}
R(\theta_0, \theta_v, \Delta\phi) = & R_{bb}(\theta_0, \theta_v, \Delta\phi) \\
& + T_{bb}^{\downarrow}(\theta_0)\rho_{bb}(\theta_0, \theta_v, \Delta\phi)T_{bb}^{\uparrow}(\theta_v) + T_{bd}^{\downarrow}(\theta_0)\rho_{db}(\theta_v)T_{bb}^{\uparrow}(\theta_v) \\
& + \left[ T_{bb}^{\downarrow}(\theta_0)\rho_{bd}(\theta_0) + T_{bd}^{\downarrow}(\theta_0)\rho_{dd} \right] T_{db}^{\uparrow}(\theta_v) \left( 1 + \rho_{dd}R_{dd} + \rho_{dd}^2 R_{dd}^2 + ... \right) \\
& + \left[ T_{bb}^{\downarrow}(\theta_0)\rho_{bd}(\theta_0) + T_{bd}^{\downarrow}(\theta_0)\rho_{dd} \right] R_{dd}\rho_{db}(\theta_v)T_{bb}^{\uparrow}(\theta_v) \left( 1 + \rho_{dd}R_{dd} + \rho_{dd}^2 R_{dd}^2 + ... \right).
\end{aligned}
\tag{36}
$$

This can then be further simplified, using the appropriate series limit, to give

$$
\begin{aligned}
R(\theta_0, \theta_v, \Delta\phi) = & R_{bb}(\theta_0, \theta_v, \Delta\phi) \\
& + T_{bb}^{\downarrow}(\theta_0)\rho_{bb}(\theta_0, \theta_v, \Delta\phi)T_{bb}^{\uparrow}(\theta_v) + T_{bd}^{\downarrow}(\theta_0)\rho_{db}(\theta_v)T_{bb}^{\uparrow}(\theta_v) \\
& + \frac{\left[ T_{bb}^{\downarrow}(\theta_0)\rho_{bd}(\theta_0) + T_{bd}^{\downarrow}(\theta_0)\rho_{dd} \right] \left[ T_{db}^{\uparrow}(\theta_v) + R_{dd}\rho_{db}(\theta_v)T_{bb}^{\uparrow}(\theta_v) \right]}{1 - \rho_{dd}R_{dd}}.
\end{aligned}
\tag{37}
$$

Finally, the reflectance at the top of the cloud (TOC), including molecular absorption below the cloud, is obtained by scaling the terms in Eq. 37 by the appropriate below cloud clear-sky transmittance terms, subscripted with "bc":

$$
\begin{aligned}
R_{TOC}(\theta_0, \theta_v, \Delta\phi) = & R_{bb}(\theta_0, \theta_v, \Delta\phi) \\
& + \mathcal{T}_{bc}(\theta_0)T_{bb}^{\downarrow}(\theta_0)\rho_{bb}(\theta_0, \theta_v, \Delta\phi)T_{bb}^{\uparrow}(\theta_v)\mathcal{T}_{bc}(\theta_v) + \mathcal{T}_{bc,d}T_{bd}^{\downarrow}(\theta_0)\rho_{db}(\theta_v)T_{bb}^{\uparrow}(\theta_v)\mathcal{T}_{bc}(\theta_v) \\
& + \frac{\left[ \mathcal{T}_{bc}(\theta_0)T_{bb}^{\downarrow}(\theta_0)\rho_{bd}(\theta_0) + \mathcal{T}_{bc,d}T_{bd}^{\downarrow}(\theta_0)\rho_{dd} \right] \left[ \mathcal{T}_{bc,d}T_{db}^{\uparrow}(\theta_v) + R_{dd}\mathcal{T}_{bc,d}^2\rho_{db}(\theta_v)T_{bb}^{\uparrow}(\theta_v)\mathcal{T}_{bc}(\theta_v) \right]}{1 - \rho_{dd}R_{dd}\mathcal{T}_{bc,d}^2},
\end{aligned}
\tag{38}
$$

and the reflectance at TOA including molecular absorption above the cloud is obtained by scaling $R_{TOC}(\theta_0, \theta_v, \Delta\phi)$ by the appropriate above cloud clear-sky transmittance terms, subscripted with "ac":

$$
R_{TOA}(\theta_0, \theta_v, \Delta\phi) = \mathcal{T}_{ac}(\theta_0)\mathcal{T}_{ac}(\theta_v)R_{TOC}(\theta_0, \theta_v, \Delta\phi),
\tag{39}
$$

where $\mathcal{T}_{ac}(\theta) = \mathcal{T}_{ac}(0, p_c)^{\sec\theta}$ and $\mathcal{T}_{bc}(\theta) = \mathcal{T}_{bc}(0, p_c)^{\sec\theta}$, and $\mathcal{T}_{ac}(0, p_c)$ and $\mathcal{T}_{bc}(0, p_c)$ are above cloud and below cloud nadir transmittances, respectively, interpolated from transmittance profiles $\mathcal{T}_{ac}(0, p)$ and $\mathcal{T}_{bc}(0, p)$, respectively, obtained from RTTOV as described in section 3.1. For diffuse radiation the nadir transmittances are scaled in a manor appropriate for a isotropic radiation field. Consider an isotropic radiation field $L(\theta, \phi) = L$ incident on a layer of optical depth $\tau$. The incident irradiance is

$$
E^i = \int_0^{2\pi} \int_0^{\pi/2} L(\theta, \phi) \sin\theta \cos\theta d\theta d\phi = \pi L
\tag{40}
$$





20  and the transmitted irradiance neglecting Rayleigh scattering (accounted for in the cloud operators) is

$$E^{\mathrm{t}} = \int_0^{2\pi} \int_0^{\pi/2} L(\theta,\phi) e^{-\tau/\cos\theta} \sin\theta \cos\theta d\theta d\phi \tag{41}$$

$$= 2\pi L \int_0^{\pi/2} e^{-\tau/\cos\theta} \sin\theta \cos\theta d\theta. \tag{42}$$

One can equate this to transmission through a scaled optical depth $\tau^*$ so

$$\pi L e^{-\tau^*} = \int_0^{2\pi} \int_0^{\pi/2} L(\theta,\phi) e^{-\tau/\cos\theta} \sin\theta \cos\theta d\theta d\phi, \tag{43}$$

$$e^{-\tau^*} = 2 \int_0^{\pi/2} e^{-\tau/\cos\theta} \sin\theta \cos\theta d\theta, \tag{44}$$

$$\mathcal{T}_{\mathrm{d}} = 2 \int_0^{\pi/2} \mathcal{T}(\theta) \sin\theta \cos\theta d\theta. \tag{45}$$

### 3.6.2  Thermal brightness temperature

The observed TOA brightness temperature is given by (assuming dependence on $\lambda$)

$$L_{\mathrm{TOA}}(\theta_{\mathrm{v}}) = L_{\mathrm{ac}}^{\uparrow} + \left[ L_{\mathrm{ac}}^{\downarrow} R_{\mathrm{db}}^{\uparrow}(\theta_{\mathrm{v}}) + B(T_{\mathrm{c}})\varepsilon(\theta_{\mathrm{v}}) + L_{\mathrm{bc}}^{\uparrow} T_{\mathrm{db}}^{\uparrow}(\theta_{\mathrm{v}}) \right] \mathcal{T}_{\mathrm{ac}}(\theta_{\mathrm{v}}), \tag{46}$$

where $L_{\mathrm{ac}}(\theta_{\mathrm{v}})$ is the upward radiance into the viewing direction from the atmosphere above the cloud, $L_{\mathrm{ac}}^{\downarrow}$ is the downward radiance from the atmosphere above the cloud, $L_{\mathrm{bc}}^{\uparrow}$ is the upward radiance from the atmosphere below the cloud, $B(T_{\mathrm{c}})$ is the Planck function as a function of the cloud top temperature $T_{\mathrm{c}}$ and $\varepsilon(\theta_{\mathrm{v}})$ is the cloud emissivity operator described in section 3.4. The clear-sky radiance terms $L$ are obtained from RTTOV as described in section 3.1.

Since $L_{\mathrm{bc}}^{\uparrow}$ is computed as a preprocessing task and is therefore static, it depends on the surface temperature $T_{\mathrm{s}}$ which is a retrieved parameter. Therefore, $L_{\mathrm{bc}}^{\uparrow}$ must be updated during the retrieval process. For this we approximate $L_{\mathrm{bc}}^{\uparrow}$ by assuming a linear relationship between $T_{\mathrm{s}}$ and $L_{\mathrm{bc}}^{\uparrow}$ around the a priori base state, i.e. the ancillary surface temperature input, given as

$$L_{\mathrm{bc}}^{\uparrow} = L_{\mathrm{bc,a}}^{\uparrow} + (T_{\mathrm{s}} - T_{\mathrm{s,a}}) \frac{\partial L_{\mathrm{bc,a}}^{\uparrow}}{\partial T_{\mathrm{s,a}}} \mathcal{T}_{\mathrm{bc,d}}, \tag{47}$$

where $L_{\mathrm{bc,a}}^{\uparrow}$ is the upward radiance from the atmosphere below the cloud computed with the a priori surface temperature $T_{\mathrm{s,a}}$.

## 3.7  Derivatives

The gradient of the forward model $(\partial y/\partial x)$, where $y$ is a radiance measurement in a single channel and $x$ is one of the retrieved parameters, is required for the following two purposes:



1. The gradient with respect to parameters which are to be derived from the measurements (state parameters) is required for the inversion of the non-linear forward model discussed in section 4.

2. The gradient with respect to parameters which are considered known and are not retrieved, e.g. meteorology, surface reflectance and surface emissivity, is used to judge the sensitivity to these parameters and thus to estimate their contribution to the retrieval uncertainty.

Derivatives of the forward model may be obtained through straightforward linearization of the forward model equations already given and, as a result, the derivations will not be presented here.

## 3.8 Assumptions and limitations

The forward model introduces some assumptions and limitations that contribute to uncertainty and may under certain conditions bias retrieval results. Inaccuracies which result from these assumptions and limitations are termed forward model uncertainty and do not include uncertainties in the input atmospheric and surface parameters (termed retrieval parameter uncertainty) or uncertainties in RTTOV (these are forward model uncertainty, but their evaluation lies outside the scope of this paper). The assumptions and limitations may be grouped into two lists. The first list involves limitations related to instrument resolution and assumptions related to limited information content:

– Satellite pixels are assumed to be either completely clear or completely overcast. Retrievals from pixels with subgrid variability, i.e. broken cloudiness, will be biased and therefore unrepresentative of the clouds within the pixel.

– Satellite pixels are assumed to be either completely of land or completely of ocean so that the BRDF and emissivity assumptions will be either for land or ocean. Retrievals from pixels with both land and ocean, such as with coastlines, islands and inland waters, will be biased since the BRDF and emissivity will be unrepresentative of at least part of the pixel.

– The liquid water droplet size distribution has an assumed shape and width and only varies in effective radius. Deviations from this assumption will result in biases particularly in optical thickness and effective radius. Although it is theoretically possible to retrieve additional size distribution parameters besides effective radius, the lack of information in typical multispectral image measurements has made this impractical in the current implementation of the forward model.

– Ice crystal scattering properties are computed based on shape (habit) and size. The ice cloud bulk scattering models used in the forward model must assume a size distribution and a mixture of possible habits. These assumptions are based on in-depth analysis of aircraft-based in-situ measurements. Deviations from this assumption will result in biases, particularly in optical thickness and effective radius.

The second list involves assumptions and limitations in the radiative transfer solution:





- The forward model characterizes the cloud layer with an infinitely thin geometric thickness. Since the peak sensitivity of the thermal channels to the cloud is within the cloud itself, the cloud will be placed at a height below the top of a real cloud with finite geometric thickness.

- The forward model contains only a single cloud layer. Retrievals from pixels with more than one cloud layer, where the upper cloud layer is optically thin (cirrus overlying liquid water cloud), will be the result of radiance contributions from both clouds resulting in a bias away from the properties of either cloud. For example, relative to the cirrus cloud and assuming typical cloud conditions, the optical thickness will be biased high, effective radius will be biased low, and the cloud top pressure will be at a level between the cloud layers.

- Each pixel is processed independently, which means that the radiative transport that occurs in each pixel occurs independently of that in the neighbouring pixels, i.e. horizontal transport between neighbouring pixels is not accounted for. In the literature this is referred to as the independent pixel approximation (Cahalan et al., 1994). It has been shown that the biases incurred can be significant at higher spatial resolutions whereas on the order of 1 km the subgrid variability mentioned above will usually dominate (Heidinger and Stephens, 2002).

- The forward model is a scalar operator model in which radiation that is incident and reflected/transmitted from/through the cloud layer or reflected from the surface is modeled as one of two types: directional or hemispherical. This is in contrast to multi-stream models that model reflection and transmission with a range of quadrature points over the upward and downward hemispheres. The use of scalar operators is one of the primary reasons the forward model is orders of magnitude faster than a multi-stream model. Note that the operators themselves are computed with a multi-stream model to accurately account for multiple scattering effects within the cloud layer.

- Finally, the assumption of a plane-parallel atmosphere and ignoring polarization effects, both discussed in section 3.4, should also be included in this list.

### 3.9 Validation

In this section we discuss the evaluation of the forward model with respect to a "reference forward model". The focus will be based on the use of scalar operators versus a multi-stream solution while all other aspects of the two models are essentially

identical, i.e. they use the same input parameters, both use RTTOV for gas transmittance, both use the same method to compute Rayleigh scattering parameters and both have the same limitations and assumptions listed in the first list of section 3.8. The assumptions of a geometrically infinitely thin cloud and on ignoring polarization effects are common and well explored in the literature. Likewise, the plane-parallel assumption and the assumption of 1-D geometry are inherent in both the fast forward model and the reference model on which numerous studies have been performed (see Heidinger and Stephens (2002) and

authors therein). The assumption of a single layer in a multilayer case is covered in a another study performed by the authors (McGarragh et al., 2017a).





The reference forward model divides the atmosphere into as many layers as the meteorological input contains. Gas transmittance is taken from the clear-sky RTTOV computations. The Rayleigh scattering optical thickness for dry air $\tau_{\mathrm{Ray}}(\lambda)$ is computed according to Justus and Paris (1985). The Rayleigh scattering phase matrix $F_{\mathrm{Ray}}(\lambda, \Theta)$ is computed from the depo-

larization ratio which may be determined from the King factor for dry air (Peck and Reeder, 1972). Liquid water particle single scattering properties are computed using the Mie theory code implementation presented by Grainger et al. (2004) assuming the same size distribution parameters used for the fast forward model. The refractive indices for liquid water are taken from Hale and Querry (1973). Ice crystal single scattering properties are taken from the same source as the fast forward model. A standard vector discrete ordinate solution is performed accounting for absorption, emission and multiple scattering with

solar and thermal sources using XRTM (McGarragh, 2017). Unlike the fast forward model, for channels with both a solar and thermal component a single solution is performed. In addition, the solution includes delta-M scaling (Wiscombe, 1977), the TMS single scattering correction of Nakajima and Tanaka (1988) and the so-called pseudo spherical approximation in which the solar beam is modelled through a spherical shell (Dahlback and Stamnes, 1991).

The comparisons that follow are presented in the form of 2-D plots of fractional difference given by

$$\Delta(x) = \frac{x_{\mathrm{ref}} - x_{\mathrm{fast}}}{x_{\mathrm{ref}}}, \tag{48}$$

where $x$ is either reflectance $R$ or brightness temperature $L$. The comparisons revolve around a base state where any two parameters in the state are varied in a plot. The base state can be summarized as follows:

- Mid-latitude summer temperature, pressure and trace gas profiles provided by McClatchey et al. (1972).

- All four BRDF operators, $\rho_{\mathrm{bb}}(\lambda, \theta_0, \theta_{\mathrm{v}}, \Delta\phi)$, $\rho_{\mathrm{bd}}(\lambda, \theta_0)$, $\rho_{\mathrm{db}}(\lambda, \theta_{\mathrm{v}})$ and $\rho_{\mathrm{dd}}(\lambda)$, are set to 0.2.

- Surface emissivity $\epsilon_{\mathrm{s}}(\lambda) = 0.8$.

- Retrieval parameters $\tau_{\mathrm{c}}$ and $r_{\mathrm{e,c}}$ are set to the a priori values indicated in Tables 5 and 6 for liquid water and ice cloud, respectively.

- Retrieval parameter $p_{\mathrm{c}}$ is set to 800 hPa and 245 hPa for liquid water and ice cloud, respectively.

- Retrieval parameter $T_{\mathrm{s}}$ is set to 290° K.

- Solar zenith angle $\theta_0 = 35.0°$, satellite zenith angle $\theta_{\mathrm{v}} = 35.0°$ and relative azimuth angle $\Delta\phi = 90.0°$.

The results are given as a function of optical thickness versus effective radius, solar zenith angle, relative azimuth angle and the four BRDF parameters for 0.65 μm; and effective radius, cloud top pressure and surface temperature for 3.70 μm and 11.0 μm. The satellite zenith angle is omitted as the results for it are relatively similar to those for the solar zenith angle. For the BRDF parameters $\rho_{\mathrm{dv}}$ and $\rho_{\mathrm{dd}}$ are combined as they both account for incident diffuse radiation and the effect of $\rho_{\mathrm{dv}}$ is

15 usually relatively small.

We provide the minimum and maximum values ($\Delta_{\min}$ and $\Delta_{\max}$, respectively) in order to maintain a useful scale in the plots. In addition, $\Delta$ values equivalent to pre-launch measurement noise requirements for the comparable MODIS channels





are also provided as $|\Delta_{\mathrm{req}}|$. MODIS is an instrument with relatively high accuracy requirements and one of the Cloud_cci instruments. This value by no means should be considered a requirement for the difference between the two models and is only there as a relevant reference. The difference between these models is quite a different value especially since we assume that the reference model, although more accurate than the fast model, surely cannot simulate the measurements exactly.

In Figure 3 fractional differences between the reference and fast forward models are presented for liquid water cloud at 0.65, 3.70 and 11.0 µm. The choice of the wavelengths is sufficient to cover the full range of wavelengths used for our retrieval, i.e. a shortwave solar wavelength where multiple scattering of solar radiation will dominate (0.65 µm), a thermal wavelength dominated by thermal emission and absorption (11.0 µm) and a mixed channel with both a solar and a thermal component (3.70 µm). Note that values in the red direction are due to overestimation by the fast forward model relative to the reference forward model and values in the blue direction are due to underestimation. At 0.65 µm it is apparent that the errors tend to be small above an optical thickness of approximately 10. Small in this case is considered close to $|\Delta_{\mathrm{req}}|$. One exception is at a solar zenith angle of around 20°, corresponding to single scattering angles of around 140° where there is rapid variation from the first and second rainbows from 120–150°. The other two exceptions are at relative azimuth angles of around 90°, a single scattering angle of around 132° (again within the rainbow region) and at 180°, where at the base state solar and satellite zenith angles of 35° corresponds to backscattering. It is also apparent that the differences tend to be low below an optical thickness of approximately 0.1. This indicates that the surface reflection is well characterized in the solution since most of the signal at low optical thicknesses will be from the surface. It is in the critical region of optical thickness around unity – the transition between single and multiple scattering – where the differences are largest. For variation in bidirectional reflectance there is a sweet spot around 0.2–0.3 with overestimation below and underestimation above. This is due the fact that the model does not account for the reflection from the bottom of the cloud of incident beam radiation from the surface and the sweet spot falls where this is compensated by incident diffuse radiation. The same minimum in the differences occurs for $\rho_{\mathrm{0d}}$ and $\rho_{\mathrm{dv}} + \rho_{\mathrm{dd}}$, although to a lesser degree.

For the thermal wavelengths the differences remain below 0.5% for variation in effective radius and cloud top pressure. Interestingly, as a function surface temperature the difference can be much larger for a optical thicknesses below 10 away from the base state of 290 K. This is due to the linear approximation of Eq. 47.

For ice cloud (Figure 4) the results are, in almost all cases, better than that for liquid water. Most of the variation as a function of effective radius has disappeared, most likely due to the flatter phase function for ice particles relative to liquid water droplets. For the same reason, most of the features in the plots as a function of solar zenith and relative azimuth angle have also disappeared. It is also apparent that, as a function of the BRDF parameters, the underestimation has grown somewhat at the expense of overestimating a change attributed to the flatter phase function and a slight change in the balance discussed above.





## 4  Retrieval method

### 4.1  Optimal estimation

The ORAC retrieval algorithm is based on the optimal estimation approach for atmospheric inverse problems described by
Rodgers (2000) in which the input state to a forward model is optimized to obtain the best match between real measurements
and simulated measurements computed with a forward model while being constrained by a priori knowledge of the state. The
relationship between the $n$ element state vector $\boldsymbol{x}$ and the $m$ element measurement vector $\boldsymbol{y}$ is given by

$$\boldsymbol{y} = \mathbf{F}(\boldsymbol{x}, \boldsymbol{b}) + \boldsymbol{\epsilon}, \tag{49}$$

where $\mathbf{F}$ is the forward model, $\boldsymbol{b}$ is the set of all other assumed model parameters not in the state vector $\boldsymbol{x}$ and $\boldsymbol{\epsilon}$ represents
the measurement and forward model error. Optimal estimation falls in the category of statistical inversion methods based on
Bayes' theorem:

$$P(\boldsymbol{x}|\boldsymbol{y}) = \frac{P(\boldsymbol{y}|\boldsymbol{x})P(\boldsymbol{x})}{P(\boldsymbol{y})}, \tag{50}$$

where $\boldsymbol{x}$ and $\boldsymbol{y}$ are continuous random variables, $P(\boldsymbol{x})$ is the prior probability density function (PDF) of the state $\boldsymbol{x}$ before
the measurements are made, $P(\boldsymbol{y})$ is the prior PDF of the measurements $\boldsymbol{y}$ before the measurements are made, $P(\boldsymbol{y}|\boldsymbol{x})$ is the
conditional PDF of $\boldsymbol{y}$ given $\boldsymbol{x}$ and $P(\boldsymbol{x}|\boldsymbol{y})$ is the conditional PDF of $\boldsymbol{x}$ given $\boldsymbol{y}$. The solution is obtained by minimizing $P(\boldsymbol{x}|\boldsymbol{y})$
to obtain the maximum posteriori solution, the solution that has the maximum probability of being the truth. If the PDFs are
assumed to follow a Gaussian distribution $P(\boldsymbol{x}|\boldsymbol{y})$ can be expressed as a $\chi^2$ distribution:

$$\chi^2 = -2\ln P(\boldsymbol{x}|\boldsymbol{y}) = [\boldsymbol{y} - \mathbf{F}(\boldsymbol{x}, \boldsymbol{b})]^{\mathrm{T}} \mathbf{S}_\epsilon^{-1} [\boldsymbol{y} - \mathbf{F}(\boldsymbol{x}, \boldsymbol{b})] + (\boldsymbol{x} - \boldsymbol{x}_{\mathrm{a}})^{\mathrm{T}} \mathbf{S}_{\mathrm{a}}^{-1} (\boldsymbol{x} - \boldsymbol{x}_{\mathrm{a}}), \tag{51}$$

where $\mathbf{S}_\epsilon$ is the measurement, forward model and forward model parameter error covariance matrix, $\boldsymbol{x}_{\mathrm{a}}$ is the *a priori* state
vector and $\mathbf{S}_{\mathrm{a}}$ is the *a priori* error covariance matrix. $\boldsymbol{x}_{\mathrm{a}}$ and $\mathbf{S}_{\mathrm{a}}$ denote the best guess of the state before the measurement
is made and the uncertainty of this guess, respectively. Eq. 51, known as the cost function, is a combination of the squared
deviations between the measurements and the forward model and the retrieved state vector and the a priori state vector, each
weighted by their associated covariance matrix. The retrieval problem is that of finding the minimum value of $\chi^2$.

As with most atmospheric inverse problems, our cloud retrieval problem is ill-posed in that noise in the measurements $\boldsymbol{y}$
and forward model/parameter errors leads to significant errors in the estimate of $\boldsymbol{x}$. A problem is well posed if for any $\boldsymbol{y}$ a
solution $\boldsymbol{x}$ exists, the solution of $\boldsymbol{x}$ is unique and the solution is stable with respect to perturbations in $\boldsymbol{y}$ (Engl et al., 2000;
Vogel, 2002). If any of these conditions are not met then the problem is ill-posed leading to non-existence, non-uniqueness (due
to discretization of the problem) and/or ill-conditioning (due to amplification of errors in $\boldsymbol{x}$ due to errors in $\boldsymbol{y}$) (Doicu et al.,
2010). It is for this reason that an a priori constraint is required. The fact that the problem is non-linear requires an iterative
method. Finally, in order to perform the iteration efficiently, while maintaining a stable step size, a form of regularization is
required.





In ORAC regularization is achieved with the Levenberg-Marquardt (Levenberg, 1944; Marquardt, 1963) method applied to
Gauss-Newton iteration leading to

$$\boldsymbol{x}_{i+1} = \boldsymbol{x}_i + \left(\mathbf{S}_\mathrm{a}^{-1} + \mathbf{K}_i^\mathrm{T}\mathbf{S}_{\epsilon,i}^{-1}\mathbf{K}_i + \gamma_i\mathbf{D}_i\right)^{-1}\left\{\mathbf{K}_i^\mathrm{T}\mathbf{S}_{\epsilon,i}^{-1}\left[\boldsymbol{y} - \mathbf{F}(\boldsymbol{x}_i,\boldsymbol{b})\right] - \mathbf{S}_\mathrm{a}^{-1}\left(\boldsymbol{x}_i - \boldsymbol{x}_\mathrm{a}\right)\right\}, \tag{52}$$

where the subscript $i$ denotes the number of the current iteration, $\mathbf{K}_i$ is the $m \times n$ weighting function matrix, $\gamma_i$ is the Levenberg-Marquardt regularization parameter and $\mathbf{D}_i$ is an $n \times n$ diagonal scaling matrix. Each column of $\mathbf{K}_i$ contains the derivatives of the forward model with respect to each state parameter given by

$$k_{i,j,k} = \frac{\partial f_j(\boldsymbol{x}_i,\boldsymbol{b})}{\partial x_k}. \tag{53}$$

Thus, for a linear system, we could write $\boldsymbol{y} = \mathbf{K}_i(\boldsymbol{x}_i - \boldsymbol{x}_0)$, where $\boldsymbol{x}_0$ is some reference state.

Central to the Levenberg-Marquardt method is the regularization parameter $\gamma_i$, which controls the type of step taken at each iteration. On the one hand, if $\gamma_i \to 0$, the algorithm behaves like Gauss-Newton iteration, which will provide an exact solution to a linear problem in one iteration. On the other hand, if $\gamma_i \to \infty$, the step direction tends to steepest descent and the step size tends to zero. The optimal value of $\gamma_i$ will be one that maximally reduces the cost function for each iteration. The procedure for determining the value of $\gamma_i$ is to start with a small value (so the initial iteration will resemble Gauss-Newton), then at each iteration:

- If, as a result of the step given by Eq. 52, the cost function increases, do not update the state vector and increase $\gamma_i$ for the next step.

- If the cost function is decreased by the step given by Eq. 52, update the state vector and decrease $\gamma_i$ for the next step.

Our implementation uses a factor of 10 for increasing and decreasing $\gamma_i$. The initial value $\gamma_0$ is chosen to be the mean of the diagonal elements of the Hessian $\mathbf{K}_i^\mathrm{T}\mathbf{S}_{\epsilon,i}^{-1}\mathbf{K}_i$.

The scaling matrix $\mathbf{D}$ is used to ensure that the state space parameters are of similar magnitude to avoid ill-conditioned matrix operations and to therefore ensure numerical stability. Alternatively, in ORAC $\mathbf{D} = \mathbf{I}$, where $\mathbf{I}$ is the identity matrix, and the scaling is performed directly on the state vector parameters, their a priori values, their a priori uncertainties, and the derivatives of the simulated measurements w.r.t. the state vector parameters according to the scaling values in Tables 3 and 4 for liquid water and ice cloud, respectively. With this method, scaling parameters may be chosen in a more intuitive way than setting the scaling matrix $\mathbf{D}$ directly. Finally, in some cases the step size will be large enough to push state parameters out of a physically reasonable range. In this case the values are bound to the ranges listed in Tables 3 and 4.

This iterative procedure, presented as a flow chart in Figure 5, is continued until either the convergence criteria are satisfied, or a maximum number of iterations is exceeded. In the former case the retrieval is said to have "converged" while the latter case can generally be rejected as a failed retrieval. ORAC uses the change in the cost function between iterations to determine whether the algorithm has converged. A negligible change in cost between iterations indicates that the retrieval is no longer improving the fit between the measurements and the forward model. In ORAC the cost change threshold is set to $0.05m$ and the maximum number of iterations is set by "trial and error" to 40, although the number of iterations usually required to





**Table 3.** Liquid water cloud scaling parameters and lower and upper retrieval limits.

| Parameter | $\log_{10}(\tau_{0.55,c})$ | $r_{e,c}$ (µm) | $p_c$ (hPa) | $T_s$ (K) |
| --- | --- | --- | --- | --- |
| Scaling | 10.0 | 1.0 | 1.0 | 1.0 |
| Lower limit | -3.0 | 0.1 | 10.0 | 250.0 |
| Upper limit | 2.408 | 35.0 | 1200.0 | 320.0 |

**Table 4.** Ice cloud scaling parameters and lower and upper retrieval limits.

| Parameter | $\log_{10}(\tau_{0.55,c})$ | $r_{e,c}$ (µm) | $p_c$ (hPa) | $T_s$ (K) |
| --- | --- | --- | --- | --- |
| Scaling | 10.0 | 1.0 | 1.0 | 1.0 |
| Lower limit | -3.0 | 0.1 | 10.0 | 250.0 |
| Upper limit | 2.408 | 100.0 | 1200.0 | 320.0 |

achieve convergence is less than half this value. An additional, purely Gauss-Newton step ($\gamma = 0$) is performed to test for false convergence. If this step changes the cost by greater than one then $\gamma$ is reinitialized and the iteration continues.

After successful convergence the retrieved state $\hat{x}$ is set to $x_i$, where $i$ is the index of the last iteration, and an uncertainty estimate for the retrieved state $\hat{S}$ is calculated with

$$\hat{S} = \left( S_a^{-1} + K_i^T S_{\epsilon,i}^{-1} K_i \right)^{-1}, \tag{54}$$

where the uncertainty of a particular parameter $\hat{x}_k$ is defined as the square root of the corresponding diagonal element $\sigma_k = \sqrt{\hat{S}_{kk}}$.

### 4.2 Measurement vector and covariance matrix

In general, the measurement vector $y$ contains solar reflectance (during the day) and/or thermal brightness temperature values (day and night) for any number of channels. For the retrieval described in this paper the required measurements are those that correspond to the wavelengths of the so-called AVHRR heritage channels: 0.615/0.630 (AVHRR-2/AVHRR-3), 0.862, 1.61, 3.74, 10.8 and 12.0 µm and as a result, for the rest of this paper, we will assume that these channels are available during the day while at night only the thermal channels 3.74, 10.8 and 12.0 µm are available. (Please see the introduction of this paper and the references therein for more details on the sensitivities of these channels.) The 1.61-µm channel is only available from a subset of the AVHRR-3 platforms and, in these cases, to save bandwidth the 1.61-µm channel is used during the day while at night the 3.74-µm channel is used. These are our effective radius sensitive channels and to maintain a consistent record over the AVHRR heritage we elect to use only the 3.74 µm channel by default unless it is unavailable during the day for AVHRR in which the 1.61 µm channel is used. The other instruments used with Cloud_cci and their channel configurations are discussed in Part I.





The optimal estimation framework allows for explicit inclusion of uncertainties from the measurements, the forward parameters and the forward model itself. These uncertainties are combined into the so-called "measurement and forward model" covariance matrix $\mathbf{S}_{\epsilon,i}$ for inversion iteration $i$ given by

$$\mathbf{S}_{\epsilon,i} = \mathbf{S}_y + \mathbf{S}_{\mathrm{fm}} + \mathbf{K}_{b,i}\mathbf{S}_b\mathbf{K}_{b,i}^{\mathrm{T}}, \tag{55}$$

where $\mathbf{S}_y$ is the covariance matrix describing the measurement uncertainties, $\mathbf{S}_{\mathrm{fm}}$ describes the forward model uncertainties due

to incomplete physics or computational approximations, $\mathbf{S}_{\mathrm{b}}$ describes the uncertainties in the forward model input parameters in $\boldsymbol{b}$ and $\mathbf{K}_{b,i}$ is a weighting function matrix which propagates this uncertainty into measurement space and is given by

$$k_{b,i,j} = \frac{\partial f_i(\boldsymbol{x}_i, \boldsymbol{b})}{\partial b_j}. \tag{56}$$

The dependence of $\mathbf{S}_{\epsilon,i}$ on the iteration comes from the dependence of $\mathbf{K}_{b,i}$ on the state vector, i.e. $\mathbf{K}_{b,i}$ is a linearization around a base state defined by $\boldsymbol{x}_i$. Although it is possible to include covariance in the CC4CL implementation, in the current

configuration all three covariance matrices are assumed to be diagonal, that is, it is assumed that there is no correlation in the noise from different channels and no correlation in the forward model and forward model parameter uncertainties. $\mathbf{S}_y$ is based on the prelaunch error characterization of the instrument noise given in Part I. Uncertainties in the forward model itself $\mathbf{S}_{\mathrm{fm}}$ were determined through rigorous sensitivity studies (Watts et al., 1998; Sayer, 2009; Siddans et al., 2011) and include effects from sub pixel inhomogeneity, assumed liquid water droplet size distribution, ice crystal models (habit and size distribution), the surface BRDF model, the radiative transfer assumptions and assumptions made in the "fast" radiative transfer solution. Uncertainties in the forward model parameters $\mathbf{S}_b$ are obtained from published uncertainties, if available, and include uncertainties in meteorology (pressure, temperature, water vapour and ozone) and surface parameters such as temperature,

wind vector, surface reflectance and emissivity and ice/snow extent.

### 4.3  State vector and a priori state vector and covariance matrix

The retrieval state vector $\boldsymbol{x}$ can be written as

$$\boldsymbol{x} = \begin{bmatrix} \log_{10}(\tau_{0.55,\mathrm{c}}) \\ r_{\mathrm{e,c}} \\ p_{\mathrm{c}} \\ T_{\mathrm{s}} \end{bmatrix}, \tag{57}$$

where $\tau_{0.55,\mathrm{c}}$ is the total cloud optical thickness at a wavelength of 0.55 µm, $r_{\mathrm{e,c}}$ (µm) is the cloud particle effective radius,

$p_{\mathrm{c}}$ (hPa) is the cloud top pressure and $T_{\mathrm{s}}$ (K) is the temperature of the surface. The transformation to $\log_{10}$ space for optical thickness is desirable as the forward model is a strong non-linear function of optical thickness. An added benefit is that since optical thickness is assumed to have a normally distributed PDF the possibility of negative values will be avoided in $\log_{10}$ space. The size distribution for liquid water droplets is assumed to be the modified gamma distribution as discussed in section 3.2.1 and aircraft measurements of ice crystals from optical probes indicate exponential type distributions (Heymsfield





**Table 5.** Liquid water cloud a priori values and associated uncertainties.

| Parameter | $\tau_{0.55,c}$ | $r_{e,c}$ (µm) | $p_c$ (hPa) | $T_s$ (K) (sea/land) |
|---|---|---|---|---|
| A priori value | 6.3 | 12.0 | 900.0 | - |
| A priori uncert. | $1 \times 10^8$ | $1 \times 10^8$ | $1 \times 10^8$ | 2.0/5.0 |

**Table 6.** Ice cloud a priori values and associated uncertainties.

| Parameter | $\tau_{0.55,c}$ | $r_{e,c}$ (µm) | $p_c$ (hPa) | $T_s$ (K) (sea/land) |
|---|---|---|---|---|
| A priori value | 6.3 | 30.0 | 400.0 | - |
| A priori uncert. | $1 \times 10^8$ | $1 \times 10^8$ | $1 \times 10^8$ | 2.0/5.0 |

and Platt, 1984; Arnott et al., 1994; Mitchell and Arnott, 1994; Kinne et al., 1997; Wyser, 1998), for neither of which is a $\log_{10}$ transformation of the effective radius appropriate. The relationship of the forward model with cloud top pressure and surface temperature is weakly non-linear therefore these parameters are also retrieved as absolute values.

The a priori state vector $\boldsymbol{x}_a$ is written analogously to the state vector $\boldsymbol{x}$ and the a priori covariance matrix $\mathbf{S}_a$ is diagonal, i.e. the a priori standard deviations are assumed to be uncorrelated. The a priori state vector depends on the assumed phase, the values for which are presented in Tables 5 and 6, along with the their associated uncertainties for liquid and ice cloud, respectively. The values for the cloud parameters, $\tau_{0.55,c}$, $r_{e,c}$ and $p_c$ are chosen to be typical average values for each phase, although in the current retrieval setup the standard deviation for these values is set to $10^8$, so that the corresponding diagonal elements of $\mathbf{S}_a$ are $10^{16}$, effectively eliminating any constraint they have on the retrieval.

The choice of surface temperature $T_s$ as a retrieval parameter is so that information in thermal channels can be used to refine the assumed surface temperature, minimizing errors which could bias retrieval results, especially cloud top pressure. Retrievals will be particularly sensitive to surface temperature in the case of thin clouds due to significant contributions to the measured brightness temperature in the thermal channels from both the cloud and the surface. For example, assuming a negative lapse rate, a positive change in surface temperature will have a similar effect as a positive change in cloud top pressure. In the case of low clouds, the thermal contrast between the surface and the clouds will be smaller approaching, the uncertainties of thermal measurements and the surface temperature. Due to the similar, approximately linear, relationship that thermal measurements have with cloud top pressure and surface temperature, some constraint is required. Since cloud top pressure is unconstrained, the surface temperature must be constrained. The a priori surface temperature is obtained from the ECMWF ERA-Interim reanalysis input and is given standard deviations of 5 and 2 K for land and sea, respectively, so that the corresponding diagonal element of $\mathbf{S}_a$ is either 25 or 4. Comparison with in-situ measurements made at the surface indicate the errors in the reanalysis





surface temperature on the average remain well within these uncertainties but can approach and or exceed these values in situations such as ocean upwelling near the land and over land surfaces such as desert with strong diurnal effects. The values of 5 and 2 K for land and sea have been chosen through trial and error to provide an optimal balance between constraint and the use of the available measurement information on surface temperature. It should be noted that the retrieved surface temperature under cloudy conditions is not intended for use as a product but that we expect it to at least have some reduction in uncertainty

relative to that of the ECMWF reanalysis inputs.

It should be noted that the estimate of uncertainty does not account for systematic errors in the a priori, in particular on a regional basis. Even if the a priori inputs are unbiased globally they will have some regional bias. Users should be aware that when averaging this data, the uncertainty will not tend towards zero as the a priori uncertainty is systematic.

Our retrieval algorithm has different pathways depending on illumination conditions: "day" (solar zenith angle $\theta_0 < 80°$),

"twilight" ($80 \leq \theta_0 < 90°$) or "night" ($\theta_0 \geq 90°$). During the day all solar and thermal channels provided are used and the state vector is complete. During twilight conditions, due to the difficulty of modelling solar radiation at solar zenith angles greater than $80°$, we use only channels with a thermal component (11 and 12 μm) and include only $p_c$ and $T_s$ in the state vector. This can lead to significant biases as the cloud optical thickness and effective radius, affecting cloud infrared transparency, are fixed at the a priori values. As a result, twilight retrievals should be used with caution, especially for ice cloud. At night (no

solar signal) the retrieval is limited to the thermal channels 3.7, 11 and 12 μm and lacks a large amount of optical thickness and effective radius information but even in the thermal-only measurements a significant amount of this information on these parameters exists (see the section 1). As such, we include these parameters in the state vector for night (as in day) to allow these to vary in a way that is consistent with thermal-only measurements. This significantly improves our estimate of cloud top pressure at night. Ultimately, we do not report these in the final product at night but together they contain enough information for a single microphysical parameter so-called "effective emissivity" discussed in section 4.6. Effective emissivity is a common parameter tied to cloud top pressure retrievals (see the section 1) used to describe cloud transparency that can be used for nighttime radiative flux calculations.

## 4.4  First guess

The first guess $x_0$ defines the state of the inversion iteration at $i = 0$. This is distinct from the a priori state $x_a$ which is the best estimate of the state *before* the measurements are made, i.e. it is independent of the measurements, such as that from climatology or the reanalysis input. It is common to use the a priori state as the first guess but this is not the only option. In fact, measurements may be used to determine a first guess that is closer to the retrieval than the a priori resulting in faster

convergence of the inversion. In some cases the choice of first guess may change the retrieval result depending on the existence of less than optimal minima that may or may not exist. In our case, the first guess for $\tau_{0.55,c}$, $r_{e,c}$ and $T_s$ are set to the a priori values.

The first guess for $p_c$ is derived from the measurements by interpolating the observed brightness temperature in the 11-μm window channel within the reanalysis temperature profile, which is in fact a simple retrieval of cloud top pressure in itself,




assuming an opaque cloud. The methodology carefully deals with both tropospheric inversions and the tropopause to bypass
their effect. The steps involved are:

1. Remove inversions including the tropopause from the temperature profile.

   (a) Skip past any surface inversion by searching for the lowest level at which the temperature decreases with height.

   (b) Locate inversions within the boundary layer defined as levels with a temperature lower than that of the level above
20        them.

      i. If an inversion is found, locate the top of the inversion, being the next level up at which the temperature
         decreases relative to the previous.

      ii. Overwrite all values from the bottom of the inversion to two points above the top of the inversion (assuring the
         inversion has some width) by linearly extrapolating from the two levels just beneath the inversion.

(c) Locate the tropopause as the lowest level between 500 and 30 hPa for which the lapse rate is less than 2 K km$^{-1}$
       and remains below that level for at least 2 km.

   (d) Overwrite all values from the tropopause up by extrapolating from the two levels just beneath the tropopause.

2. Interpolate the 11-µm brightness temperature onto the new temperature profile.

   (a) If the brightness temperature is outside the range of the profile set $p_{c,0}$ to the minimum/maximum temperature (as
30        appropriate).

   (b) Search through the profile for the first pair of levels that bound the requested temperature. For a liquid phase
       retrieval search from the bottom of the profile up. Otherwise, search top down.

   (c) Linearly interpolate brightness temperature between those located levels to determine $p_{c,0}$.

## 4.5 Diagnostics

The retrieval implementation produces a number of diagnostic fields, a few of which will be mentioned here as they are
presented later. First, the final cost $\chi^2$ (Eq. 51) normalized by the number of measurements $m$, given by $\chi_N^2 = \chi^2/m$, is output.
This quantity serves to indicate how well the measurements fit the forward model given the final estimated state vector, i.e. the
state vector from the last iteration. A value of less than unity is generally accepted as a "good fit", but it should be noted that a
good fit does not necessarily mean that the retrieval is an accurate estimate of the true state. We will show in section 5, that due
to non-uniqueness in the retrieval space, it is possible to obtain a good fit with an unrepresentative estimate of the state. The
number of iterations used to achieve convergence on the retrieved state is also output. This can also be useful to indicate the
quality of the retrieval but, again, a large/small number of iterations does not necessarily indicate a poor/good retrieval. Finally,
two information quantities are produced including the averaging kernel matrix $\mathbf{A}$ and the number of degrees of freedom for
signal $d_s$. The averaging kernel is given by

$$\mathbf{A} = \frac{\partial \hat{\boldsymbol{x}}}{\partial \boldsymbol{x}} = \mathbf{GK} = (\mathbf{K}^T \mathbf{S}_\epsilon^{-1} \mathbf{K} + \mathbf{S}_a^{-1})^{-1} \mathbf{K}^T \mathbf{S}_\epsilon^{-1} \mathbf{K}, \tag{58}$$



where $\mathbf{G}$ is referred to as the gain matrix. $\mathbf{A}$ quantifies the response of the retrieval to changes in the true state vector about the retrieved state vector. The diagonal elements of $\mathbf{A}$ range from zero to one, where for a perfect retrieval $\mathbf{A}$ would be an identity matrix indicating that changes in each state vector element are perfectly represented by the retrieval. Instead of the averaging kernel we will present the degrees of freedom for signal given by

$$d_{\mathrm{s}} = \mathrm{tr}(\mathbf{A}), \tag{59}$$

where $\mathrm{tr}(\mathbf{A})$ is the trace of $\mathbf{A}$, which describes the number of useful independent quantities there are in the measurements, i.e. the amount of independent pieces of information.

### 4.6 Derived products

Several products are produced that are derived from the retrieved state and the assumed input parameters. These include cloud
top height $H_{\mathrm{c}}$ (km), cloud top temperature $T_{\mathrm{c}}$ (K), cloud water path (CWP, g m$^{-2}$), spectral cloud albedo $R_{\mathrm{bd}}(\lambda, \theta_0)$ and spectral cloud effective emissivity $\varepsilon(\theta_{\mathrm{v}})$.

$H_{\mathrm{c}}$ and $T_{\mathrm{c}}$ are obtained by linear interpolation onto the pressure profile at the retrieved cloud top pressure $p_{\mathrm{c}}$.

CWP is derived from the retrieved cloud optical thickness $\tau_{0.55,\mathrm{c}}$ and effective radius $r_{\mathrm{e,c}}$ (Han et al., 1994) with

$$\mathrm{CWP} = \frac{4}{3} \frac{\tau_{\mathrm{c},0.55} r_{\mathrm{e,c}} \rho}{Q_{\mathrm{e}}}, \tag{60}$$

assuming a density $\rho$ for water and ice of 1.0 and 0.9167 g cm$^{-3}$, respectively, and an extinction coefficient $Q_{\mathrm{e}}$ for water and ice of 2.0 and 2.1 g cm$^{-3}$, both valid for particles large with respect to wavelength.

A "spectral cloud albedo", a directional-hemispheric reflectance, also referred to as *black-sky albedo*, defined as (with dependence on $\lambda$, $\tau_{\mathrm{c},0.55}$ and $r_{\mathrm{e,c}}$ assumed)

$$
\begin{aligned}
R_{\mathrm{bd}}(\theta_0) &= \frac{\int_0^{2\pi} \int_0^{\pi/2} R_{\mathrm{bb}}(\theta_0, \theta_{\mathrm{v}}, \Delta\phi) \cos\theta_{\mathrm{v}} \sin\theta_{\mathrm{v}} \, d\theta_{\mathrm{v}} \, d\Delta\phi}{\int_0^{2\pi} \int_0^{\pi/2} \cos\theta_{\mathrm{v}} \sin\theta_{\mathrm{v}} d\theta_{\mathrm{v}} d\Delta\phi} \\
&= \frac{1}{\pi} \int\limits_0^{2\pi} \int\limits_0^{\pi/2} R_{\mathrm{bb}}(\theta_0, \theta_{\mathrm{v}}, \Delta\phi) \cos\theta_{\mathrm{v}} \sin\theta_{\mathrm{v}} \, d\theta_{\mathrm{v}} \, d\Delta\phi,
\end{aligned} \tag{61}
$$

is derived from the retrieved $\tau_{\mathrm{c},0.55}$ and $r_{\mathrm{e,c}}$, where $R_{\mathrm{bb}}(\lambda, \tau_{\mathrm{c}}, r_{\mathrm{e,c}}, \theta_0, \theta_{\mathrm{v}}, \Delta\phi)$ is the bidirectional reflectance of either liquid water or ice cloud introduced in section 3.4. Cloud albedo $R_{\mathrm{bd}}(\lambda, \tau_{\mathrm{c},0.55}, r_{\mathrm{e,c}}, \theta_0)$, is obtained from LUTs, depending on
phase and wavelength, built in the same manner as the operators introduced in section 3.4. Similarly "spectral cloud effective emissivity" $\varepsilon(\lambda, \tau_{\mathrm{c},0.55}, r_{\mathrm{e,c}}, \theta_{\mathrm{v}})$ is derived from the LUTs and is in fact the same as the cloud emissivity operator described in section 3.4. At night, this parameter can be thought of as containing the microphysical information retrieved in the optical thickness and effective radius, both of which are not reported at night.

Finally, to account for the fact that the retrieval algorithm may place the cloud top at a location lower than the physical
top as discussed in section 3.8, depending on the optical thickness of the cloud, cloud top pressure, height, and temperature corrections are performed as presented in McGarragh et al. (2017b). The result of this correction is made available as separate





outputs from the non-corrected versions (i.e. for comparison to CALIOP observations). It must be noted that the result of this correction will no longer be radiatively consistent with the other retrieved variables. In other words, broadband radiative flux computations using the corrected cloud top pressure values will be biased.

As with the primary retrieval parameters, an estimate of the uncertainty in the derived parameters is computed. For this, standard uncertainty propagation is used

$$\sigma_{x_{\mathrm{d}}} = \sqrt{\sum_{i=1}^{n}\sum_{j=1}^{n} \hat{S}_{ij} \frac{\partial x_{\mathrm{d}}}{\partial x_i} \frac{\partial x_{\mathrm{d}}}{\partial x_j}}, \tag{62}$$

where $x_{\mathrm{d}}$ is a particular derived parameter.

## 5   Retrieval performance

In this section we test the performance of the retrieval system just presented. The methodology can be summarized as follows:

1. First, produce radiances using the "fast" forward model for a given set of parameters. These parameters include the parameters that are assumed to be known, such as meteorology and surface reflectance/emissivity to which a Gaussian variability is applied in accordance with their uncertainty. The parameters also include the set of retrieval parameters which are taken as exact.

2. Then, apply Gaussian noise to the radiances. For this we choose the prelaunch noise characteristics for MODIS.

3. Finally, perform a retrieval on the simulated radiances and compare the results to the retrieval parameters used to produce

the radiances.

The channels simulated and subsequently used in the retrieval were the MODIS bands comparable to the AVHRR heritage channels 0.630 0.862, 3.74, 10.8 and 12.0 μm, specifically MODIS bands 1, 2, 20, 31 and 32. Note that we performed the same performance analysis using the 1.61 μm channel rather than the 3.74 μm channel and the results were similar.

   In the comparisons that follow we look at fractional error of the retrieved optical thickness $\tau_{\mathrm{c},0.55}$, effective radius $r_{e,\mathrm{c}}$ and

cloud top pressure $p_{\mathrm{c}}$ defined as

$$\epsilon(x) = \frac{x - \hat{x}}{x}, \tag{63}$$

where $x$ is the true value of a retrieved parameter and $\hat{x}$ is the retrieved estimate of the parameter. In addition, we will show the estimate of the uncertainty of each retrieval parameter normalized by the retrieved estimate given by

$$\sigma(x) = \frac{\sigma_x}{\hat{x}}, \tag{64}$$

where $\sigma(x)$ is the estimated standard deviation of the parameter $x$ such that $\sigma_{x_k} = \sqrt{\hat{S}_{kk}}$, where $k$ is the index of the parameter in the state vector $\mathbf{x}$. Finally, we will show the final cost $\chi^2$ normalized by the number of measurements $m$: $\chi^2_{\mathrm{N}}$, the number of iterations used and the degrees of freedom for signal computed for the retrieved state $d_{\mathrm{s}}$.





Our base state describes the parameters that are not explicitly indicated as something else in our discussion of the results. These include the use of the same base state values used for the forward model validation in section 3.9. Unlike the forward model validation, in the interest brevity, we do not show results as a function of geometry and choose the same base state values ($\theta_0 = 35.0°$, $\theta_v = 35.0°$ and $\Delta\phi = 90.0°$).

In Figure 6 we present daytime liquid water cloud retrieval results as a function of optical thickness $\tau_c$ and effective radius $r_{c,e}$, with everything else set to our base state. A cost $\chi^2 < 1$ is considered a good fit. It is clear that there is a good fit in all the retrievals. This does not necessarily mean the retrievals are accurate, only that a good fit has been obtained. As already mentioned, these retrievals suffer from non-uniqueness which means it is possible to obtain a good fit for a less than optimal solution. Looking at the error $\epsilon(x)$ (red/blue represent overestimation/underestimation) for optical thickness and effective radius we can see that there is a breakdown around the critical optical thickness of unity where a transition from the single scattering regime to multiple scattering approximately occurs. It is worth pointing out that absolute errors in optical thickness are still quite small at low cloud optical thickness in the context of cloud retrievals. Given the measurement information available it is difficult to obtain an accurate retrieval for optical thicknesses less than 0.1. We show these low optical thicknesses since subvisible cirrus (optical depth < 0.03) are known to exist (Jensen et al., 1996; Reverdy et al., 2012) but also to demonstrate that the retrieval is acceptable most cases (optical depth > 1.0). There is also an increase in the number of iterations and degrees of freedom at low optical thicknesses. The increase in the degrees of freedom is due to the increased sensitivity to surface temperature at low cloud optical thicknesses making it more difficult to distinguish the thermal cloud signal from the thermal surface signal therefore subjecting the retrieval to more non-uniqueness. The increases in the error in effective radius at low optical thicknesses and smaller particle sizes is due to a decreased sensitivity to smaller particles. With cloud top pressure there is a very small error. It should be noted that the analysis does not take into account the potential biases from treating the cloud as geometrically infinitely thin, as discussed in section 3.8, since we used that model to produce our synthetic measurements. The uncertainties for optical thickness are below 20% for optical thicknesses from about 4 to 30 but increase outside of this range and are invariant with effective radius. For effective radius the uncertainties are almost all below 20% above an optical thickness of approximately 5 except for effective radii below approximately 6 μm. Finally, the uncertainties for cloud top pressure are all below approximately 10% for an optical thickness greater than unity.

For ice cloud (Figure 7) the cost increases slightly for a small set of retrievals, the number of iterations increases for an optical thickness around unity and the degrees of freedom for optical thicknesses less than unity also increase relative to liquid water due to a stronger sensitivity to ice cloud at thermal wavelengths and therefore to surface emission. The error for optical thickness is similar to that for liquid water while the larger error in effective radius when $\tau_c < 1$ is now for larger particles and opposite in sign. This suggests that the peak in sensitivity to effective radius is at around 30 μm and the effective radius is subject to overestimation for smaller particles and underestimation for larger particles. For cloud top pressure, the error is larger for ice cloud when $\tau_c < 1$, which is most likely due to an increase in the importance of absorption and emission for larger particles at thermal wavelengths which will decrease in importance as optical thickness increases from the single scattering regime to multiple scattering. Finally, relative to liquid water cloud, the minimum in the optical thickness uncertainties shifts to slightly lower optical thicknesses, the effective radius uncertainties are improved down to an optical thickness of around





0.5, due to greater sensitivity to effective radius at thermal wavelengths, and the cloud top pressure uncertainties are somewhat larger due to the reason stated above for error and due to the greater sensitivity to surface emission and the difficulty of
separating cloud and surface signals.

Figures 8 and 9 repeat Figures 6 and 7, respectively, except at night. Even though we do not currently report optical thickness and effective radius at night, we still present them here to show that there is enough sensitivity to these parameters to help the cloud top pressure retrieval. From the cost we see that we obtain a good fit for liquid water cloud and the number of iterations is mostly lower than 14. The degrees of freedom varies little, with values around three. The reason for this is due to the
lack of sensitivity to optical thickness and therefore relatively little change to the sensitivity to surface temperature with optical thickness. The error in optical thickness is beyond 50% almost everywhere except for some parts of the 1–10 μm effective radius region. The error for effective radius is significantly better and it is this microphysical information that really helps improve the cloud top pressure retrieval of liquid water cloud, which has very little error for clouds of greater than approximately 1 optical depth. The uncertainties for optical thickness are all greater than 100% while for effective radius they are lower for
larger optical thicknesses and effective radii (30–70%), as expected. For cloud top pressure the uncertainties are almost all less than 20% for optical thicknesses greater than unity.

For ice cloud (Figure 9) the night retrieval obtains good fits for all, a slightly larger number of iterations in the optical thickness region around unity and similar degrees of freedom compared to the liquid water cloud retrieval at night. The interesting observation is that the retrieval of optical thickness at night is better for ice cloud in the range between 1 and 10. This is
important as it is in this range that semi-transparent cirrus clouds occur. Without this sensitivity to optical thickness, the cloud top pressure would be significantly over estimated. Notice that at optical thicknesses below unity, both the optical thickness and cloud top pressure are over estimated. Finally, like the rest of the cases, estimates of uncertainty are over estimated but one can argue the estimate at night is better for ice cloud than for liquid water cloud. The uncertainties for optical thickness and effective radius are improved relative to liquid water cloud in cases where the error is also improved and the uncertainties for
cloud top pressure are mostly still below 20% at optical thicknesses greater than unity except at large effective radii.

In Figure 10 we show daytime liquid water cloud retrieval results as a function of optical thickness $\tau_c$ and cloud top pressure $p_c$, with otherwise everything else set to our base state. We perform the retrieval for both a relatively small (for liquid water droplets) effective radius ($r_{e,c} = 4$ μm) and a relatively large effective radius ($r_{e,c} = 20$ μm). We choose these values to be well away from the a priori effective radius for liquid water cloud of 12 μm. The small effective radius retrieval suffers from
10 large cost values at optical thicknesses from 1–10 and at cloud top pressure values lower than approximately 550 hPa, albeit most liquid water clouds will be lower in the atmosphere. The number of iterations is consistent with the cost, with more iterations used for higher cost retrievals. The larger degrees of freedom at large cloud top pressure values, even with larger optical thicknesses, is due to the closer proximity to the surface, making it harder to distinguish the cloud from the surface thermal signals. The retrieval errors are mostly what we expect for liquid water cloud. For optical thickness, the errors small
for lower clouds with an optical thickness greater than approximately unity (bearing in mind that most liquid water clouds will have considerably larger optical thicknesses than unity). For effective radius small errors begin at an optical thickness of approximately two. Finally, for cloud top pressure the error is small for optical thicknesses greater than unity for clouds below





400 hPa. For an effective radius of 20 μm the results are in general better than for 4 μm. This is not surprising as the sensitivity shifts closer to 3.7 μm and the other thermal channels.

Figure 11 shows the same results as Figure 10 but for ice cloud. In this case we choose small and large effective radii of $r_{e,c} = 10$ μm and $r_{e,c} = 50$ μm, respectively, to be well away from the a priori effective radius for ice cloud of 30 μm. The results are, in general, better as larger errors are confined to lower optical thicknesses. The cost and number of iteration plots are much better in the region of large values that were observed in the corresponding plots for liquid water. As we have seen before, for most of the cases of optical thickness less than unity, the degrees of freedom for ice increases relative to liquid

water. Also, unlike for the small effective radius, the degrees of freedom is larger for optical thicknesses less than unity at all levels, not just low levels. This is due to the greater sensitivity to larger particles at thermal wavelengths and the difficulty of separating the cloud and surface emission signals. Finally, looking at the effective radius errors for both the liquid water cloud with an effective radius of 4 (Figure 10) and the effective radius errors for ice cloud with an effective radius of 50 (Figure 11) we can see the flip in errors that we discussed in reference to Figures 6 and 7 for small/large liquid water/ice particles.

In Figure 12 we show daytime liquid water cloud retrieval results as a function of optical thickness $\tau_c$ and $r_{c,e}$. In this case we perform the retrieval for both a relatively small surface albedo ($A = 0.01$) and a relatively large albedo ($A = 0.9$), with everything else set to our base state. For an albedo of 0.01 almost all the cost results indicate good fits, with only a small spike around an optical thickness of unity for small effective radii, while the number of iterations are all reasonable. The degrees of freedom follows the same pattern as in Figure 6, which is the same as this case except with the base state albedo of 0.2, but the values at low optical thicknesses are much higher. This is because of the increased influence of the surface emission signal if we assume the surface emissivity is one minus albedo. The errors are almost all acceptable (even for cloud top pressure at small optical thicknesses). These results indicate why aerosol retrievals generally perform well over ocean and that the retrieval of subvisible cirrus properties may also be possible over ocean. For an albedo of 0.9 the results for optical

thickness and effective radius are significantly different and highlight the difficulty of retrieving these properties over bright surfaces (bearing in mind that an albedo of 0.9 is typical of fresh snow cover). The degrees of freedom is now fixed at three, the value we would expect if surface temperature has little effect. The retrieval of optical thickness is overestimated for optical thicknesses less than unity, something that we expect as the surface will look like cloud in this case. More interesting is that optical thickness is underestimated for optical thicknesses greater than ten. Our first thought would be that at these optical

thicknesses the surface should have little effect and that this error suggests a problem with non-uniqueness and that we may be getting stuck in a suboptimal minimum. The retrieval of effective radius in this case is also problematic, with underestimation at small optical thicknesses and over estimation at large values. In fact, the opposite direction of the errors compared to that of optical thickness further suggests a problem with non-uniqueness.

    For ice cloud (Figure 13) the results with the low and high albedos are comparable to that of liquid water, with the differences

primarily with the large cost values for large optical thicknesses and large effective radii and the over estimation of cloud top pressure for optical thicknesses less than unity. In this case it is the overestimation of optical thickness that drives the cloud top pressure solution lower in the atmosphere.





# 6   Conclusions

This paper describes the optimal estimation component of the Community Cloud retrieval for Climate (CC4CL) based on the

Optimal Retrieval of Aerosol and Cloud (ORAC) algorithm. An extensive forward model is described which includes emission, absorption and multiple scattering of radiation from both solar and thermal sources. The surface is characterized by a BRDF specific to either ocean or land. The model's "fast" radiative transfer solution is separated into solar and thermal components and the model's assumptions and limitations were addressed. Validation was undertaken with a reference forward model, i.e. a more extensive forward model that attempts to eliminate some of the most important assumptions in the "fast" solution.

Results show that, in relation to the simple scalar operators, for optical thicknesses greater than 10, the errors are comparable to instrument noise, but it should be noted that this error is the difference between the reference forward model and the "fast" forward model and not a measure of the total errors in forward modeling. At small optical thicknesses (less than 0.1–1.0) the errors become larger, especially at optical thicknesses approaching the critical regime of unity, where the contribution of single and multiple scattering to the total shortwave signal are comparable. Fortunately, these optical thicknesses of less than unity

are uncommon for most cases in cloud remote sensing.

The retrieval method is then described including the optimal estimation approach, the input measurements and a priori quantities along with their associated uncertainties, our choice of the iteration first guess and quantities derived from the retrieved parameters. Particular attention was focused on the estimation of the retrieval uncertainty. The performance of the retrieval method was assessed theoretically by simulating measurements using a range of values for the retrieval parameters and then subsequently performing a retrieval on these simulated measurements to which Gaussian noise levels as appropriate

for MODIS were added. The errors are less than 10% for optical thicknesses larger than 10 and less than 20% for optical thicknesses larger than unity. For night our retrieval does not report cloud optical thickness or effective radius but uses the information content in these values to improve the cloud top pressure results. These results are consistent with our forward model analysis. For optical thicknesses less than unity the results become problematic which could have implications for the retrieval of subvisible cirrus but, as with successful aerosol retrievals over dark surfaces (such as ocean), the results are

comparable to those of optical thicknesses larger than 10. Finally, compared to the actual errors our estimation of the retrieval uncertainty is comparable, again, at cloud optical thickness greater than unity.

It is worth noting that the ORAC algorithm is being extended to retrieve properties in two cloud layers. The publication for the work is in progress and will have the approximate citation of McGarragh et al. (2017a).

*Code availability.*   ORAC/CC4CL is free and open source software and is licensed under the GNU General Public License version 3. It can

be downloaded at: https://github.com/ORAC_CC/ORAC.

*Acknowledgements.*   This study was funded as part of NERC's support of the National Centre for Earth Observation. This work was supported by the European Space Agency through the Cloud_cci project (contract: 4000109870/13/I-NB).



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





**Figure 3.** Fractional differences $\Delta(x)$ between the reference forward model and the fast forward model for liquid water cloud as a function of optical thickness $\tau_{0.55,c}$ versus effective radius $r_{e,c}$ (µm), solar zenith angle $\theta_0$, relative azimuth angle $\Delta\phi$, bidirectional surface reflectance $\rho_{0v}$, directional-diffuse surface reflectance $\rho_{0d}$, and a combination of diffuse-directional and diffuse-diffuse surface reflectance $\rho_{dv} + \rho_{dd}$ for wavelength $\lambda = 0.65$ and effective radius $r_{e,c}$ (µm), cloud top pressure $p_c$ (hPa) and surface temperature $T_s$ (K) for 3.70 and 11.00 µm.





**Figure 4.** Same as Figure 3 but for ice cloud.





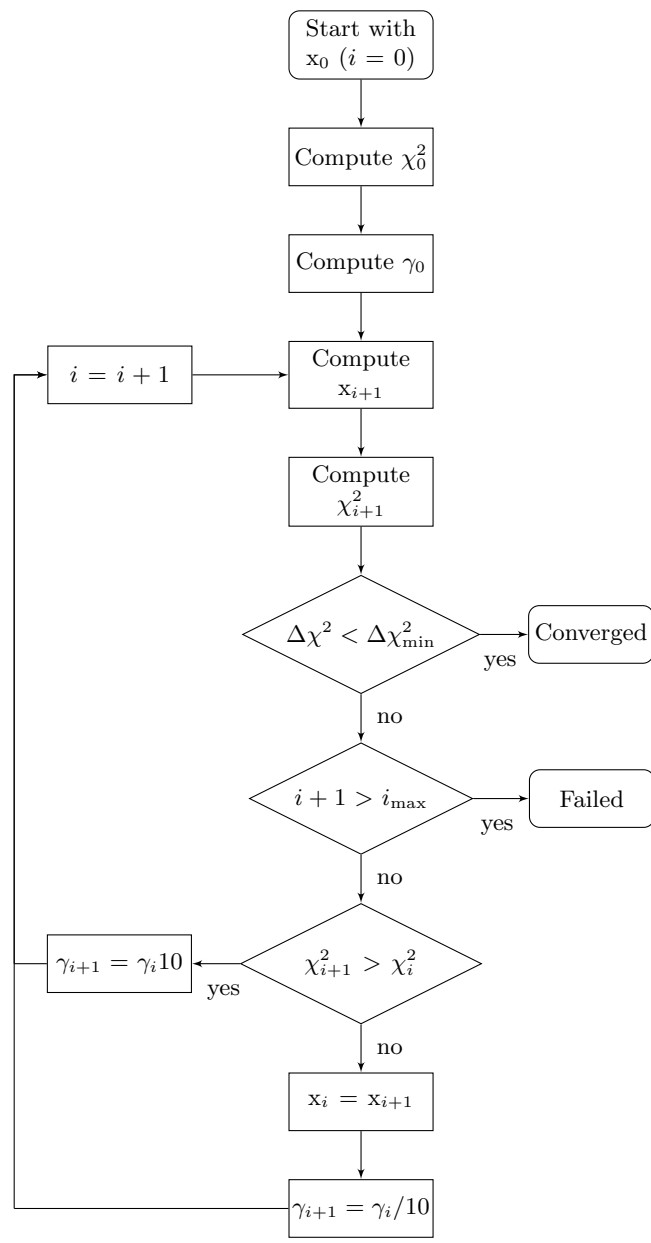

**Figure 5.** The ORAC inversion system depicted as a flow chart. Rectangles with corners are processes, diamonds are decisions and rectangles with rounded corners are start and top terminals.





**Figure 6.** Daytime liquid water cloud retrieval results as a function of optical thickness $\tau_c$ and effective radius $r_{c,e}$ with everything else set to the base state. The top row presents metrics of retrieval performance, the middle row the fractional error $\epsilon$, and the bottom row shows the fraction uncertainty.



**Figure 7.** Same as Figure 6 but for ice cloud.



**Figure 8.** Same as Figure 6 but for the night retrieval path.





**Figure 9.** Same as Figure 7 but for the night retrieval path.





**Figure 10.** Daytime liquid water cloud retrieval results as a function of optical thickness $\tau_c$ and cloud top pressure $p_c$ for an effective radius of 4 μm (top) and 20 μm (bottom) otherwise everything else is set to the base state.





**Figure 11.** Same as Figure 10 but for ice cloud.







**Figure 12.** Daytime liquid water cloud retrieval results as a function of optical thickness $\tau_c$ and effective radius $r_{c,e}$ for a surface albedo of 0.01 (top) and 0.9 (bottom) otherwise everything else is set to the base state.



**Figure 13.** Same as Figure 12 but for ice cloud.