# Peer review of "The Community Cloud retrieval for CLimate (CC4CL). Part II: The optimal estimation approach"

_Atmospheric Measurement Techniques, 2017_

## Referee Comment (RC1) · Anonymous Referee #1 · 16 Nov 2017

This paper describes the optimal estimation algorithm for the retrieval of cloud top pressure (CTP), optical thickness (COT), and effective particle radius (CER), namely the Optimal Retrieval of Aerosol and Cloud (ORAC) algorithm, used in ESA's Community Cloud retrieval for Climate (CC4CL) system. The theoretical basis and methodology of the retrieval are described in great detail. Also presented is an evaluation of the retrieval itself and the fast forward radiative transfer model, each using simulated TOA radiances under a variety of assumptions.

This is a very detailed paper, and is generally well written. I do think revisions are necessary, however. My detailed comments are below. Note that there appear to be

line number issues within the available pdf, e.g., line 10 towards the bottom of pages – I try to remain consistent with what appears in the file. Also, there appear to be font inconsistencies within Figs. 6-13 that caused missing text labels in my downloaded version (viewing online was fine) – please check these figures for portability.

Comments

p 3, line 5: I'm not sure this statement on CO2-slicing is correct. It is my understanding that you need at least two channels within the 13-15$\mu$m region, as the technique relies on ratios of two channels having differential CO2 absorption. It is indeed not applicable to the AVHRR record, though, as the authors correctly state.

p 3, lines 20-22: Do the authors have references showing that this difference in simulated vs observed radiances is indeed the case? For the solar retrievals, as long as the thermal cloud top is used consistently throughout the retrieval (e.g., for above-cloud atmospheric corrections), then any solar radiance bias due to using the thermal cloud top is essentially "built in" to the retrievals themselves.

p 3, lines 33-35: It's not clear to my why other "traditional" algorithms cannot also be applied in the same consistent manner? Can the authors elaborate?

p 4, lines 5-6: This statement is true for many "traditional" retrieval algorithms as well; for instance, the Nakajima-King approach will produce simultaneous COT and CER retrievals that are radiatively consistent over both wavelengths used in the inversion.

p 4, lines 7-8: Yes, this may be an advantage of optimal estimation generally speaking, but it is immaterial here since active sensors are not used.

p 4, lines 13-14: This is not unique to optimal estimation – see, e.g., the MOD06 retrievals which use mathematics analogous to optimal estimation to calculate retrieval uncertainty estimates.

p 4, lines 15-16: This is certainly true!

p 4, lines 17-19: The authors might want to add the Wang et al. papers (2016a,b, JGR, doi:10.1002/2015JD024528, doi:10.1002/2015JD024526) that detail a MODIS IR-based optimal estimation retrieval of ice phase COT, CER, and CTH.

Section 3.2.2: Did the authors verify that the Baum general habit mixture for ice clouds yields consistent COT retrievals from independent solar and thermal IR retrievals? When combining observations from different spectra, it is important to use assumptions that can provide consistency between both. Refer to the Holz et al. (2016, ACP, doi:10.5194/acp-16-5075-2016) paper for further discussion.

p 11, lines 9-11: Why extend the retrieval space beyond the $60\mu$m maximum CER of the ice cloud scattering properties? What does this gain?

p 12, line 23: Remove "look-up" from the beginning of this line, as that is already included in the acronym LUT.

p 12-13, DISORT approximation list: I would also add that the analytical solution requires approximating the scattering phase function, through some use of expansion terms, in which case some special scattering angles (e.g., liquid cloud rainbow region, glory) may not be well resolved depending on the number of terms used.

p 13, lines 11-13: Can the authors provide a statement on any errors incurred by fixing the cloud top for above/below-cloud Rayleigh scattering, given that cloud top is part of the state vector?

p 13, last paragraph: Can the authors provide evidence that the errors due to not accounting for the response functions are small? In some spectral regions cloud scattering properties may have significant variation with wavelength such that characterizing channel placement is critical (e.g., I believe, ice crystal absorption around $3.7\mu$m, among others). Regardless, if the calculations are done offline and stored in LUTs anyways, why not go ahead and account for instrument response functions?

p 20, line 10: Lbc should also depend on surface emissivity, correct?

p 21, lines 23-24: To clarify, the Jacobians are calculated via mathematical formulations rather than, say, perturbing each parameter and re-calculating the radiances?

p 21, lines 32-33: Perhaps reference Zhang et al. 2012 (JGR, doi:10.1029/2012JD017655) or another of his papers here.

Section 3.8, p 22: I again might add the representation of the phase function in the radiative transfer calculations, which can introduce large errors at special scattering angles if not adequately resolved. Backscatter peak calculations are also known to be deficient in some cases, e.g., the Yang et al. ice crystal database – see Zhou and Yang (2015, Opt Express, doi: 10.1364/OE.23.011995).

p 23, line 23: Since you're including Hale & Querry here, you might as well add the Palmer & Williams and Downing & Williams references as well.

Section 3.9: Why not use DISORT as the reference model, since that is what is used for the cloud calculations? This would eliminate potential differences in the radiative transfer solution due to different model approaches. Why are errors with respect to angle space not shown for $3.7\mu$m, which has a large solar component?

Tables 3, 4: I notice that the CER limits for liquid and ice retrievals, with the exception of the maximum liquid value, are outside of the pre-calculated cloud LUT ranges. How can that be the case?

p 28, lines 25-26: In many cases assuming uncorrelated errors is inappropriate. Are there plans to include error correlation in the future? I would suggest the authors refer to Wang et al. (2016a, JGR, doi:10.1002/2015JD024528) for a detailed treatment of error covariance matrices in the IR.

p 28, line 12 (bottom of page): Is COT really normally distributed? See, e.g., Fig. 15 in King et al. (2012, IEEE TGRS doi:10.1109/TGRS.2012.2227333) – here it seems this assumption is more valid for CER.

p 28, line 13 – p 29, line 16: I think the authors are confusing individual crystal sizes

within an observed size distribution with the effective radius that is more of a "bulk" quantity.

p 29, line 19: Are uncorrelated a priori standard deviations appropriate given non-orthogonal solution spaces (e.g., COT-CER space at small optical thickness)?

p 29, lines 21-23: Assuming large a priori standard deviation, thereby eliminating its constraint on the retrieval solution, is somewhat at odds with an earlier statement in the paper implying that a priori assumptions are important (p 25, last paragraph).

p 30, lines 19-20: Did the authors verify that including Ts in the state vector reduces its uncertainty relative to ECMWF?

Section 4.4: I'm curious how close the first-guess CTP is to the final CTP solution. I assume relatively close for optically thick clouds. Have the authors looked at this?

p 34, lines 36-37: I'm surprised to see small CTP errors at small COT, though I guess the difference between the surface and cloud temperatures are small enough that surface contamination does not significantly bias the results.

Section 5: Since Ts is fixed in the simulations but is part of the state vector, have the authors looked at differences between the retrieved Ts and its fixed value? I'm wondering if this is playing a role in some of the differences observed here.

p 35, lines 27-28: Did the authors verify that the microphysical information does in fact improve the CTP retrievals? An easy test might be to simply remove CER from the state vector.
* * *

---

## Referee Comment (RC2) · Anonymous Referee #3 · 12 Jan 2018

The manuscript "The Community Cloud retrieval for CLimate (CC4CL). Part II: The optimal estimation approach" by McGarragh et al. describes in details the optimal estimation retrieval of cloud optical thickness, effective radius and cloud top pressure based on the Optimal Retrieval of Aerosol and Cloud (ORAC) algorithm. The retrieval method is presented, uncertainties and limitations are discussed. This manuscript is well-structured and well-written and provides detailed description on the optical processes and the model used. I believe those results can make some important implications for understanding of the process and application and implementation for other satellite sensor. I recommend this paper can be published in ACP after addressing a few minor areas as follows.

[Figure]

1. Please add a short description on how the cloud phase (water or ice) is determined. 2. In the abstract, please, combine the statements from lines 8-9 and 11-14 to avoid the repetition. 3. P2, line 1: Please, add "characteristics of the instrument" 4. P19, line 31: Please, introduce the Stokes vector elements. 5. P11, line 15: "=" is missing. 6. P12, line 21 : which type of interpolation is used? 7. Figures 3 and 4. I suggest to use the opposite color scale (red for positive and blue for negative numbers). Please, specify $\Delta(R)$ and $\Delta(L)$ in figure caption.

Please also note the supplement to this comment:
https://www.atmos-meas-tech-discuss.net/amt-2017-333/amt-2017-333-RC2-supplement.pdf

---

## Author Comment (AC2) · 17 Feb 2018

**Response to comments by anonymous reviewer #3 on:**
**The Community Cloud retrieval for CLimate (CC4CL).**
**Part II: The optimal estimation approach**

Greg McGarragh and coauthors

February 16, 2018, 16:35

1. **RC:** Please add a short description on how the cloud phase (water or ice) is determined.

   **AR:** We stress that this paper was written to be independent of the cloud phase determination, which is described in detail in Part I. It is mentioned in section 2 that a cloud phase classification is a required input and that it is used to determine the appropriate cloud model. The reader is also referred to Part I for details on the cloud phase determination.

   **AC:** Nothing was changed.

2. **RC:** In the abstract, please, combine the statements from lines 8-9 and 11-14 to avoid the repetition.

   **AR:** In the abstract lines 8–9 and lines 11–14 describe results from two different analyses. Lines 8–9 describe results from the forward model validation. Lines 11–14 describe results from an evaluation of retrieval performance.

   **AC:** We left these lines unchanged.

3. **RC:** P2, line 1: Please, add "characteristics of the instrument".

   **AR:** We believe that "types of measurements used" covers instrument characteristics.

   **AC:** We left these lines unchanged.

4. **RC:** P19, line 31: Please, introduce the Stokes vector elements.

   **AR:** Is the reviewer referring to P9, line 31 (not P19)? If so, we feel that since this is a scalar forward model and since intensity I is introduced a few lines down, to save space we will leave introducing the whole Stokes vector out.

   **AC:** Nothing was changed.

5. **RC:** P11, line 15: "=" is missing.

   **AC:** This has been fixed.

6. **RC:** P12, line 21: Which type of interpolation is used?

   **AC:** We added "linearly".

7. **RC:** Figures 3 and 4: I suggest to use the opposite color scale (red for positive and blue for negative numbers).

   **AR:** If this was done it of course would have to be done in the retrieval simulation figures as well but the coauthors have discussed this and there was no consensus.

   **AC:** Nothing was changed.

8. **RC:** Please, specify (R) and (L) in figure caption.

   **AC:** We clarified the description of $\Delta(R)$ and $\Delta(L)$ in captions of figures 3 and 4.

---

## Author Response (AR1)

Dear Editor,

This document contains author responses to reviewer comments on the manuscript The Community Cloud retrieval for CLimate (CC4CL). Part II: The optimal estimation approach. In addition, as requested, a marked-up manuscript version showing the changes made is also included after the responses.

regards,

Greg McGarragh and coauthors

**Response to comments by anonymous reviewer #1 on:**
**The Community Cloud retrieval for CLimate (CC4CL).**
**Part II: The optimal estimation approach**

Greg McGarragh and coauthors

March 23, 2018, 16:59

- **RC:** Note that there appear to be line number issues within the available pdf, e.g., line 10 towards the bottom of pages — I try to remain consistent with what appears in the file. Also, there appear to be font inconsistencies within Figs. 6-13 that caused missing text labels in my downloaded version (viewing online was fine) — please check these figures for portability.

  **AR:** We would like to apologize to the reviewer about the line number issues and missing figure labels. These problems did not exist in the manuscript uploaded during the submission process and must be a result of a post processing step. We will pay careful attention to make sure that the final manuscript does not have these issues.

- **Page 3, Line 5:** I'm not sure this statement on CO2-slicing is correct. It is my understanding that you need at least two channels within the 13-15m region, as the technique relies on ratios of two channels having differential CO2 absorption. It is indeed not applicable to the AVHRR record, though, as the authors correctly state.

  **AR:** We agree with the reviewer's comment.

  **AC:** The reference to the SEVIRI retrieval has been removed.

- **Page 3, Lines 20–22:** Do the authors have references showing that this difference in simulated vs observed radiances is indeed the case? For the solar retrievals, as long as the thermal cloud top is used consistently throughout the retrieval (e.g., for above-cloud atmospheric corrections), then any solar radiance bias due to using the thermal cloud top is essentially "built in" to the retrievals themselves.

  **AR:** We are a bit confused about the reviewers response.

  This is true when the CTP is retrieved independently but when the parameters are retrieved simultaneously the CTP is *not* built in.

  Solar radiance bias as a function of CTP is primarily from variation in Rayleigh scattering above the cloud with CTP as the effect from Rayleigh scattering is pressure dependent. In the channels that we are using in this paper this is a rather small effect.

  The benefit of a simultaneous retrieval comes from using the solar channel information on COT and CER to provide information on thermal cloud transparency (effective emissivity) to better estimate the CTP. In addition, the information on COT and CER in the thermal channels is very CTP dependent and cannot be used for COT and CER unless CTP is retrieved simultaneously or known a priori.

  **AC:** We left these lines unchanged.

- **Page 3, Lines 33–35:** It's not clear to my why other "traditional" algorithms cannot also be applied in the same consistent manner? Can the authors elaborate?

  **AR:** We agree that this sentence was unclear and the arguments that we were trying to convey are actually covered in the other bullets.

  **AC:** We removed this bullet.

- **Page 4, Lines 5–6:** This statement is true for many "traditional" retrieval algorithms as well; for instance, the Nakajima-King approach will produce simultaneous COT and CER retrievals that are radiatively consistent over both wavelengths used in the inversion.

  **AR:** In the example that the reviewer supplies for their comment, namely the Nakajima-King approach, COT and CER are indeed retrieved simultaneously. In ORAC, these parameters are retrieved simultaneously along with CTP and surface temperature. This has benefits at night with cirrus clouds, where thermal measurement sensitivities to COT and CER are dependent on cloud temperature and therefore CTP. Although not as important during the day, this relationship will still take advantage of any independent thermal information on COT and CER.

  **AC:** We left these lines unchanged.

- **Page 4, Lines 7–8:** Yes, this may be an advantage of optimal estimation generally speaking, but it is immaterial here since active sensors are not used.

  **AR:** In this bullet we are trying to make the point that the algorithm presented is extensible which is a clear advantage with starting with optimal estimation rather than with a less universal approach. This bullet has been reworded to reflect this.

  **AC:** "It is extensible in that it is able to easily incorporate measurements from multiple sensors for synergistic retrieval algorithms, i.e. passive and active measurements, if and when those measurements become available."

- **Page 4, Lines 13–14:** This is not unique to optimal estimation — see, e.g., the MOD06 retrievals which use mathematics analogous to optimal estimation to calculate retrieval uncertainty estimates.

  **AR:** We believe that uncertainty estimation is built into the the optimal estimation framework in a more natural way than other less general methods. This bullet has been reworded to reflect this.

  **AC:** "It provides a rigorous characterization of the retrieval uncertainties, including propagation of measurement noise, the uncertainty of assumed parameters and the uncertainty in the forward model, that is inherently built into the system."

- **Page 4, Lines 17–19:** The authors might want to add the Wang et al. papers (2016a,b, JGR, doi:10.1002/2015JD024528, doi:10.1002/2015JD024526) that detail a MODIS IR-based optimal estimation retrieval of ice phase COT, CER, and CTH.

  **AC:** The references that the reviewer suggested have been added.

- **Section 3.2.2:** Did the authors verify that the Baum general habit mixture for ice clouds yields consistent COT retrievals from independent solar and thermal IR retrievals? When combining observations from different spectra, it is important to use assumptions that can provide consistency between both. Refer to the Holz et al. (2016, ACP, doi:10.5194/acp-16-5075-2016) paper for further discussion.

  **AR:** We did not verify that the Baum general habit mixture (with severe roughening) yields consistent COT retrievals from independent solar and thermal IR retrievals. We note that during the day both the solar and thermal channels are used simultaneously providing a consistency in itself. At night, since only the thermal channels are used, there is a potential for bias relative to the day

retrieval. Investigating different ice models and making ice model improvements is planned as future work.

**AC:** We left these lines unchanged.

- **Page 11, Lines 9–11:** Why extend the retrieval space beyond the 60µm maximum CER of the ice cloud scattering properties? What does this gain?

  **AR:** The reason for the extension is that ORAC can also use the ice crystal properties presented by Baran and Havemann [2004] which extend up to 92 µm as these properties are also used in climate models (OLR), which require larger sizes. In addition, ORAC can be used as a simulator and there are cases when larger sizes are desirable. Finally, naturally allowing the retrieval to extend past 60 µm can help identify forward model bugs and stability problems.

  **AC:** "ORAC currently accepts ice crystal scattering properties up to an effective radius of 92 µm. The ice crystal properties are only provided up to 60 µm and therefore linear extrapolation is used up to 92 µm. The error incurred in the approximation is minimal as the variation in the properties tends to flatten as effective radius increases and cloud ice crystal effective radii greater than 60µm are rare."

- **Page 12, Line 23:** Remove "look-up" from the beginning of this line, as that is already included in the acronym LUT.

  **AC:** This typo has been fixed.

- **Pages 12–13:** DISORT approximation list: I would also add that the analytical solution requires approximating the scattering phase function, through some use of expansion terms, in which case some special scattering angles (e.g., liquid cloud rainbow region, glory) may not be well resolved depending on the number of terms used.

  **AR:** The reviewer is referring to the fact that the expansion of the phase function in terms of Legendre polynomials is truncated at $2n$ terms, where $n$ is the number of hemispherical quadrature points and $2n$ is the so-called number of streams. The discretization that the reviewer mentions is a function of $n$. DISORT includes a delta-M scaling [Wiscombe, 1977], where the effects from the truncated part of the phase function (i.e. the forward scattering peak), resolved by the remaining expansion terms, are scaled into the extinction, and also the Nakajima-Tanaka TMS single scattering correction [Nakajima and Tanaka, 1988] where the single scatting effects using the truncated phase function are replaced by an *analytical* single scattering solution using the entire phase function expansion. We feel that the scattering effects that the reviewer mentions (e.g. rainbow regions, glory) are adequately resolved, for our purpose, with these corrections.

  **AC:** We left these lines unchanged.

- **Page 13, Lines 11–13:** Can the authors provide a statement on any errors incurred by fixing the cloud top for above/below-cloud Rayleigh scattering, given that cloud top is part of the state vector?

  **AR:** We have investigated this effect and found that it is only important, in the context of cloud retrievals, at wavelengths less than $\approx 0.65$ µm. We have added text to note this in the last paragraph of section 3.4 where this approximation is discussed in detail.

  **AC:** Added "We have investigated this effect and found that it is only important, in the context of cloud retrievals, at wavelengths less than $\approx 0.65$ µm."

- **Page 13, Last paragraph:** Can the authors provide evidence that the errors due to not accounting for the response functions are small? In some spectral regions cloud scattering properties may have

significant variation with wavelength such that characterizing channel placement is critical (e.g., I believe, ice crystal absorption around 3.7 μm, among others). Regardless, if the calculations are done offline and stored in LUTs anyways, why not go ahead and account for instrument response functions?

**AR:** We have re-evaluated this and found that reflectance, transmittance and emission operator errors can be significant in the AVHRR 3.7 μm channel. As the reviewer mentions, since this is an off-line calculation, there is really no reason not to account for the instrument response function. The code used to create the LUTs ("create_orac_lut", available on the internet with ORAC) now does this and new LUTs will be built for the Cloud_cci/CC4CL supported instruments.

Note that the version of the Cloud_cci processing referred to in Part I will not have this improvement but planned subsequent reprocessing will.

**AC:** The original lines explaining the monochromatic calculation have been removed and a paragraph explaining the convolution has been added starting at page 14, line 18:

"The operators are computed for each channel across the channel's spectral interval and then convolved with the channel's instrument response function with

$$[X * \mathrm{SRF}_i] = \int_{\lambda_1}^{\lambda_2} X(\lambda)\mathrm{SRF}_i(\lambda)d\lambda, \tag{1}$$

where $X$ is either reflectance $R$, transmittance $T$ or emissivity $\varepsilon$, SRF is the spectral response function for channel $i$, and $\lambda_1$ and $\lambda_2$ define the channel's spectral interval."

- **Page 20, Line 10:** Lbc should also depend on surface emissivity, correct?

  **AR:** To start with, we noticed that this sentence was poorly worded and revised it. With regards to the reviewer's comment, yes, it depends on surface emissivity, along with atmospheric temperature, water vapour amount, etc. The reason surface temperature is noted here is that it is a retrieved parameter from which updates to $L_{\mathrm{bc}}^{\uparrow}$ can be linearly derived. Surface emissivity is an assumed parameter.

  **AC:** We left these lines unchanged.

- **Page 21, Lines 23–24:** To clarify, the Jacobians are calculated via mathematical formulations rather than, say, perturbing each parameter and re-calculating the radiances?

  **AR:** Yes, that is right. The derivatives are computed analytically rather than by perturbation.

  **AC:** We have changed 'linearization' to 'analytic linearization' to clarify.

- **Page 21, Lines 32–33:** Perhaps reference Zhang et al. 2012 (JGR, doi:10.1029/2012JD017655) or another of his papers here.

  **AC:** The references that the reviewer suggested have been added.

- **Section 3.8, Page 22:** I again might add the representation of the phase function in the radiative transfer calculations, which can introduce large errors at special scattering angles if not adequately resolved. Backscatter peak calculations are also known to be deficient in some cases, e.g., the Yang et al. ice crystal database — see Zhou and Yang (2015, Opt Express, doi: 10.1364/OE.23.011995).

  **AR:** As with the related comment for pages 12–13, we believe that the phase function features the reviewer is referring to are resolved adequately with the Nakajima-Tanaka TMS single scattering correction [Nakajima and Tanaka, 1988].

  **AC:** We left these lines unchanged.

- **Page 23, Line 23:** Since you're including Hale & Querry here, you might as well add the Palmer & Williams and Downing & Williams references as well.

  **AR:** We agree with the reviewer on this. These other references were used just as they were used in the forward cloud model previously discussed.

  **AC:** These references have been added.

- **Section 3.9:** Why not use DISORT as the reference model, since that is what is used for the cloud calculations? This would eliminate potential differences in the radiative transfer solution due to different model approaches. Why are errors with respect to angle space not shown for 3.7m, which has a large solar component?

  **AR:** Note that this is actually an oversight and the current tests shown in the paper are actually performed using DISORT as the radiative transfer solver while the text is out of date. The original intention with using XRTM was to evaluate the scalar approximation with a vector solution and to include the pseudo spherical approximation, both of which XRTM supports similar to VLIDORT. Since these aspects were eventually not addressed in detail in the paper, as they are not related to assumptions made in the 'fast' solution relative to a standard multi-stream solution, we switched to DISORT for consistency. Note that when using a scalar solution and no pseudo spherical approximation XRTM produces virtually identical results to DISORT anyway.

  **AC:** The text has been fixed to indicate that DISORT is used and the reference has been modified as well.

  **AR:** It is true that the 3.7 $\mu$m channel contains a solar component and will be affected by solar geometry. In the interest of saving space we left tests as a function of the solar geometry out for the 3.7 $\mu$m channel since it is expected that the angular variation will affect the 0.65 $\mu$m channel more which can be considered an extreme case for this particular case.

- **Tables 3 and 4:** I notice that the CER limits for liquid and ice retrievals, with the exception of the maximum liquid value, are outside of the pre-calculated cloud LUT ranges. How can that be the case?

  **AR:** This was an error in that the CER limits were not up-to-date.

  **AC:** We have fixed these limits.

- **Page 28, Lines 25–26:** In many cases assuming uncorrelated errors is inappropriate. Are there plans to include error correlation in the future? I would suggest the authors refer to Wang et al. (2016a, JGR, doi:10.1002/2015JD024528) for a detailed treatment of error covariance matrices in the IR.

  **AR:** Yes there are plans to include uncertainty correlation in the future.

  **AC:** Added "It is planned to include uncertainty correlation in the next round of improvements."

- **Page 28, Line 12:** Is COT really normally distributed? See, e.g., Fig. 15 in King et al. (2012, IEEE TGRS doi:10.1109/TGRS.2012.2227333) — here it seems this assumption is more valid for CER.

  **AR:** We are aware that COT, realistically, is not normally distributed, but, in fact, whether this is the case or not was not really relevant with what we were trying to convey. The text has been reworded to better get to the point.

  **AC:** "An added benefit is that negative values of optical thickness, which may be encountered during the inversion process and must be bounded to a minimum value, will be avoided in $\log_{10}$ space."

- **Page 28, Line 13 - Page 29, line 16:** I think the authors are confusing individual crystal sizes within an observed size distribution with the effective radius that is more of a "bulk" quantity.

  **AR:** We are unclear about the reviewers comment. We are aware of the differences between individual ice crystal sizes and the effective radius of a distribution. We believe that it is made clear that we referring to the observed size distribution overall.

  **AC:** We left these lines unchanged.

- **Page 29, Line 19:** Are uncorrelated a priori standard deviations appropriate given nonorthogonal solution spaces (e.g., COT-CER space at small optical thickness)?

  **AR:** We understand that uncorrelated uncertainties are an assumption leading to room for improvement. As previously mentioned (reviewer comment for page 28, lines 25–26) there are plans to include uncertainty correlation in the future.

- **Page 29, Lines 21–23:** Assuming large a priori standard deviation, thereby eliminating its constraint on the retrieval solution, is somewhat at odds with an earlier statement in the paper implying that a priori assumptions are important (p 25, last paragraph).

  **AR:** We agree that this statement is at odds with the earlier statement. We would like to point out that the earlier statement is a part of a more general formulation and that the ORAC/CC4CL retrieval can easily be performed with stronger a priori constraints. It just happens that, for the parameters being retrieved, we find good results by only constraining the a priori surface temperature, as mentioned in the paper.

  **AC:** We left these lines unchanged.

- **Page 30, Lines 19–20:** Did the authors verify that including Ts in the state vector reduces its uncertainty relative to ECMWF?

  **AR:** It has been verified that the retrieval reduces the uncertainty relative to ECMWF depending on the cloud optical thickness and therefore the amount of surface signal. In cases where the cloud is optically thick (negligible surface signal) the lack of sensitivity to and the a priori constraint on surface temperature assures that the surface temperature does not diverge significantly from the ECMWF value. In fact, although not published yet, there is ongoing work to retrieve surface temperature in clear sky conditions with ORAC by replacing the cloud model with an appropriate aerosol model.

  **AC:** "It should be noted that the retrieved surface temperature under cloudy conditions is not intended for use as a product but, depending on cloud optical thickness, and therefore the surface signal, there is at least some reduction in uncertainty relative to that of the ECMWF reanalysis inputs. In cases where the cloud is optically thick (negligible surface signal) the lack of sensitivity and the a priori constraint assures that the surface temperature does not diverge significantly from the ECMWF value."

- **Section 4.4:** I'm curious how close the first-guess CTP is to the final CTP solution. I assume relatively close for optically thick clouds. Have the authors looked at this?

  **AR:** We have looked at this and the first guess CTP is indeed close to the final CTP solution for optically thick clouds with increasing divergence away from the final CTP as the cloud becomes optically thinner. Note that the first guess can be improved a bit by accounting for the atmospheric effects above the cloud [essentially correcting the observed TOA BT to be a TOC (top of cloud) BT] but this is not necessary as, of course, the optimal estimation retrieval accounts for this.

- **Page 34, Lines 36–37:** I'm surprised to see small CTP errors at small COT, though I guess the difference between the surface and cloud temperatures are small enough that surface contamination does not significantly bias the results.

  **AR:** In this case (figure 6) this is for a daytime retrieval where the information in the Vis/NIR measurements provide significant information on COT and CER. With this information, both the cloud signal and the signal from below the cloud can be accounted for and the CTP, even for optically thin clouds, can be estimated well. Notice that at night (figure 7) the retrieval of CTP at low COT is not as good as during the day due to less information on COT and CER in the thermal measurements.

- **Section 5:** Since Ts is fixed in the simulations but is part of the state vector, have the authors looked at differences between the retrieved Ts and its fixed value? I'm wondering if this is playing a role in some of the differences observed here.

  **AR:** We left surface temperature out of the retrieval performance evaluation in the interest of saving space as the paper is already a bit long and surface temperature is not one of the parameters of interest in this case. If one looks at the plots of degrees of freedom for signal it is apparent that there is usually 3-4 degrees of freedom. With the a priori constraint on surface temperature this leaves an average of a half degree of freedom that can be used to refine the surface temperature relative to that of ECMWF which is really the intended purpose of including it in the state vector.

- **Page 35, Lines 27–28:** Did the authors verify that the microphysical information does in fact improve the CTP retrievals? An easy test might be to simply remove CER from the state vector.

  **AR:** Leaving COT and/or CER out of the retrieval was tested when developing the night retrieval. Thermal cloud transparency is dependent on both COT and CER. The accuracy of retrieving either one at night with passive thermal measurements is limited but together they constitute enough information to account for thermal cloud transparency and are essential for an accurate CTP retrieval.

   **AC:** We clarified the description of $\Delta(R)$ and $\Delta(L)$ in captions of figures 3 and 4.

[revised manuscript text omitted]

$$\tau = \tau_{\text{Ray}} + \tau_c, \tag{29}$$

$$\omega = \frac{\tau_{\text{Ray}} + \omega_c \tau_c}{\tau_R + \tau_c}, \tag{30}$$

$$P(\theta) = \frac{\tau_{\text{Ray}} P_{\text{Ray}}(\theta) + \tau_c \omega_c P_c(\theta)}{\tau_{\text{Ray}} + \tau_c \omega_c}, \tag{31}$$

where the Rayleigh single scattering albedo is equal to unity. The computations are performed at optical thickness, effective radius, solar zenith angle, satellite zenith angle and relative azimuth angle vertices defined by the dimensions in Tables 1 and 2 for liquid water and ice cloud, respectively.

The reflectance and transmission operators represent the transfer of either direct beam or diffuse incoming radiation resulting in either direct beam or diffuse outgoing radiation. They are computed separately for both direct beam and diffuse incoming sources with results for both direct beam or diffuse outgoing radiation being produced simultaneously. In addition, an operator representing the emissivity of the cloud is produced by including a thermal source in the cloud layer.  A total of seven operators  are required (assuming dependence on $\tau_{0.55,c}$ and $r_{e,c}$):

**Table 2.** The dimensions of the ORAC LUTs used in the ice cloud retrieval. Note that not all LUTs are functions of all variables (for instance, atmospheric transmission terms are functions of a single zenith angle only).

[revised manuscript text omitted]

& + \left[ T^{\downarrow}_{\mathrm{bb}}(\theta_0)\rho_{\mathrm{bd}}(\theta_0) + T^{\downarrow}_{\mathrm{bd}}(\theta_0)\rho_{\mathrm{dd}} \right] T^{\uparrow}_{\mathrm{db}}(\theta_{\mathrm v}) \left( 1 + \rho_{\mathrm{dd}}R_{\mathrm{dd}} + \rho_{\mathrm{dd}}^2 R_{\mathrm{dd}}^2 + ... \right) \\
& + \left[ T^{\downarrow}_{\mathrm{bb}}(\theta_0)\rho_{\mathrm{bd}}(\theta_0) + T^{\downarrow}_{\mathrm{bd}}(\theta_0)\rho_{\mathrm{dd}} \right] R_{\mathrm{dd}}\rho_{\mathrm{db}}(\theta_{\mathrm v})T^{\uparrow}_{\mathrm{bb}}(\theta_{\mathrm v}) \left( 1 + \rho_{\mathrm{dd}}R_{\mathrm{dd}} + \rho_{\mathrm{dd}}^2 R_{\mathrm{dd}}^2 + ... \right).
\end{aligned}
\tag{37}
$$

This can then be further simplified, using the appropriate series limit, to give

$$
\begin{aligned}
R(\theta_0,\theta_{\mathrm v},\Delta\phi) =\ & R_{\mathrm{bb}}(\theta_0,\theta_{\mathrm v},\Delta\phi) \\
& + T^{\downarrow}_{\mathrm{bb}}(\theta_0)\rho_{bb}(\theta_0,\theta_{\mathrm v},\Delta\phi)T^{\uparrow}_{\mathrm{bb}}(\theta_{\mathrm v}) + T^{\downarrow}_{\mathrm{bd}}(\theta_0)\rho_{\mathrm{db}}(\theta_{\mathrm v})T^{\uparrow}_{\mathrm{bb}}(\theta_{\mathrm v}) \\
& + \frac{\left[ T^{\downarrow}_{\mathrm{bb}}(\theta_0)\rho_{\mathrm{bd}}(\theta_0) + T^{\downarrow}_{\mathrm{bd}}(\theta_0)\rho_{\mathrm{dd}} \right] \left[ T^{\uparrow}_{\mathrm{db}}(\theta_{\mathrm v}) + R_{\mathrm{dd}}\rho_{\mathrm{db}}(\theta_{\mathrm v})T^{\uparrow}_{\mathrm{bb}}(\theta_{\mathrm v}) \right]}{1 - \rho_{\mathrm{dd}}R_{\mathrm{dd}}}.
\end{aligned}
\tag{38}
$$

5   Finally, the reflectance at the top of the cloud (TOC), including molecular absorption below the cloud, is obtained by scaling the terms in Eq. 38 by the appropriate below cloud clear-sky transmittance terms, subscripted with "bc":

$$
\begin{aligned}
R_{\mathrm{TOC}}(\theta_0,\theta_{\mathrm v},\Delta\phi) =\ & R_{\mathrm{bb}}(\theta_0,\theta_{\mathrm v},\Delta\phi) \\
& + \mathcal{T}_{\mathrm{bc}}(\theta_0)T^{\downarrow}_{\mathrm{bb}}(\theta_0)\rho_{\mathrm{bb}}(\theta_0,\theta_{\mathrm v},\Delta\phi)T^{\uparrow}_{\mathrm{bb}}(\theta_{\mathrm v})\mathcal{T}_{\mathrm{bc}}(\theta_{\mathrm v}) + \mathcal{T}_{\mathrm{bc,d}}T^{\downarrow}_{\mathrm{bd}}(\theta_0)\rho_{\mathrm{db}}(\theta_{\mathrm v})T^{\uparrow}_{\mathrm{bb}}(\theta_{\mathrm v})\mathcal{T}_{\mathrm{bc}}(\theta_{\mathrm v}) \\
& + \frac{\left[ \mathcal{T}_{\mathrm{bc}}(\theta_0)T^{\downarrow}_{\mathrm{bb}}(\theta_0)\rho_{\mathrm{bd}}(\theta_0) + \mathcal{T}_{\mathrm{bc,d}}T^{\downarrow}_{\mathrm{bd}}(\theta_0)\rho_{\mathrm{dd}} \right] \left[ \mathcal{T}_{\mathrm{bc,d}}T^{\uparrow}_{\mathrm{db}}(\theta_{\mathrm v}) + R_{\mathrm{dd}}\mathcal{T}_{\mathrm{bc,d}}^2\rho_{\mathrm{db}}(\theta_{\mathrm v})T^{\uparrow}_{\mathrm{bb}}(\theta_{\mathrm v})\mathcal{T}_{\mathrm{bc}}(\theta_{\mathrm v}) \right]}{1 - \rho_{\mathrm{dd}}R_{\mathrm{dd}}\mathcal{T}_{\mathrm{bc,d}}^2},
\end{aligned}
\tag{39}
$$

and the reflectance at TOA including molecular absorption above the cloud is obtained by scaling $R_{\mathrm{TOC}}(\theta_0,\theta_{\mathrm v},\Delta\phi)$ by the appropriate above cloud clear-sky transmittance terms, subscripted with "ac":

$$
R_{\mathrm{TOA}}(\theta_0,\theta_{\mathrm v},\Delta\phi) = \mathcal{T}_{\mathrm{ac}}(\theta_0)\mathcal{T}_{\mathrm{ac}}(\theta_{\mathrm v})R_{\mathrm{TOC}}(\theta_0,\theta_{\mathrm v},\Delta\phi),
\tag{40}
$$

where $\mathcal{T}_{\mathrm{ac}}(\theta) = \mathcal{T}_{\mathrm{ac}}(0,p_{\mathrm c})^{\sec\theta}$ and $\mathcal{T}_{\mathrm{bc}}(\theta) = \mathcal{T}_{\mathrm{bc}}(0,p_{\mathrm c})^{\sec\theta}$, and $\
[revised manuscript text omitted]